# Ligand modulation of sidechain dynamics in a wild-type human GPCR

Lindsay D Clark[1,2†], Igor Dikiy[3†], Karen Chapman[1], Karin EJ Rödström[4], James Aramini[3], Michael V LeVine[5,6], George Khelashvili[5,6], Søren GF Rasmussen[4], Kevin H Gardner[3,7,8*], Daniel M Rosenbaum[1,2*]

[1]Department of Biophysics, The University of Texas Southwestern Medical Center, Dallas, United States; [2]Molecular Biophysics Graduate Program, The University of Texas Southwestern Medical Center, Dallas, United States; [3]Structural Biology Initiative, CUNY Advanced Science Research Center, New York, United States; [4]Department of Neuroscience and Pharmacology, University of Copenhagen, Copenhagen, Denmark; [5]Department of Physiology and Biophysics, Weill Cornell Medical College, New York, United States; [6]Institute for Computational Bioscience, Weill Cornell Medical College, New York, United States; [7]Department of Chemistry and Biochemistry, City College of New York, New York, United States; [8]Biochemistry, Chemistry and Biology PhD Programs, Graduate Center, City University of New York, New York, United States

*For correspondence:
Kevin.Gardner@asrc.cuny.edu
(KHG);
dan.rosenbaum@utsouthwestern.
edu (DMR)

†These authors contributed equally to this work

Competing interests: The authors declare that no competing interests exist.

**Abstract** GPCRs regulate all aspects of human physiology, and biophysical studies have deepened our understanding of GPCR conformational regulation by different ligands. Yet there is no experimental evidence for how sidechain dynamics control allosteric transitions between GPCR conformations. To address this deficit, we generated samples of a wild-type GPCR ($A_{2A}R$) that are deuterated apart from $^1H/^{13}C$ NMR probes at isoleucine $\delta1$ methyl groups, which facilitated $^1H/^{13}C$ methyl TROSY NMR measurements with opposing ligands. Our data indicate that low [$Na^+$] is required to allow large agonist-induced structural changes in $A_{2A}R$, and that patterns of sidechain dynamics substantially differ between agonist (NECA) and inverse agonist (ZM241385) bound receptors, with the inverse agonist suppressing fast ps-ns timescale motions at the G protein binding site. Our approach to GPCR NMR creates a framework for exploring how different regions of a receptor respond to different ligands or signaling proteins through modulation of fast ps-ns sidechain dynamics.
DOI: https://doi.org/10.7554/eLife.28505.001

## Introduction

Our understanding of the molecular underpinnings of GPCR function has been greatly advanced over the past two decades through a combination of X-ray crystal structures, computational simulations, and spectroscopic studies of protein dynamics. Crystals of bovine rhodopsin provided the first high-resolution picture of a GPCR's architecture (*Palczewski et al., 2000*; *Li et al., 2004*), and structures of photoactivation intermediates (*Nakamichi and Okada, 2006*; *Salom et al., 2006*) and retinal-free opsin (*Park et al., 2008*; *Scheerer et al., 2008*) further documented the structural transitions involved in rhodopsin activation. For GPCRs activated by diffusible ligands, crystal structures of the human $\beta_2$ adrenergic receptor ($\beta_2AR$) with inverse agonists (*Rosenbaum et al., 2007*; *Cherezov et al., 2007*), agonists (*Rosenbaum et al., 2011*; *Rasmussen et al., 2011a*; *Warne et al., 2011*), and bound G protein (*Rasmussen et al., 2011b*) provided a molecular basis for understanding how diffusible agonist binding can promote structural changes in a receptor to enhance

**eLife digest** Almost every aspect of the human body – from our senses to our moods – depends, in one way or another, on a large family of proteins called G-protein-coupled receptors. These receptor proteins, known as GPCRs for short, detect signals from outside the cell and trigger activity within the cell. This allows cells to gather information from their surroundings and to communicate with each other. Importantly, since GPCRs regulate many processes in the body that are involved in disease, it is perhaps unsurprising that over a third of all approved drugs target these receptors.

Like all proteins, GPCRs are long chain-like molecules with a repetitive backbone and short branches called sidechains. Each sidechain has its own chemical properties and electrical charge, which can affect how different parts of the chain interact with each other and what shape the protein can adopt. This in turn can influence how strongly a drug or other molecule can bind to a receptor protein.

Protein crystallography is one technique that has been used to better understand how the different GPCRs are built and how they work. The technique involves growing crystals from pure samples of the protein; this locks millions of copies of the protein in place and provides a snapshot of its shape. However, GPCRs – and especially their sidechains – are flexible and can adopt different shapes, which cannot be seen fully by only looking at protein crystals.

Now, Clark, Dikiy et al. used another technique called nuclear magnetic resonance spectroscopy, or NMR for short, to understand how drugs affect the fast moving sidechains within a GPCR. First, genetically modified yeast was used to create samples of a GPCR called the adenosine receptor $A_{2A}$ that were labelled with specific markers which made it easier to measure the structure and flexibility of the protein by NMR.

This approach revealed that too much sodium in the sample's solution supresses the large structural changes that occur in the $A_{2A}$ receptor when it binds to a drug. Moreover, it showed that the sidechains of several regions on the receptor move in different ways depending on whether the receptor binds to an activating drug or an inhibiting drug.

These findings lay the groundwork for understanding how the movements of sidechains help to activate or inhibit GPCRs, and will complement on-going studies using protein crystals. Moreover, the new approach to producing labelled proteins could be applied to other types of proteins that until now could not be studied with NMR due to practical limitations. In future, this may help scientists to better understand how drugs affect these proteins and to develop new treatments for a whole range of diseases.

DOI: https://doi.org/10.7554/eLife.28505.002

signaling. Subsequent crystal structures have revealed the ligand binding pockets for GPCRs of diverse function, responding to biogenic amines, purines, lipids, peptides, and proteins. While the sequences of these GPCRs and their orthosteric ligand binding pockets are highly diverse, the overall structures are remarkably similar, with Cα rmsd values between unrelated receptors in the 2–3 Å range. How these diverse ligands activate signaling by a small number of G proteins and arrestins through a common structural scaffold remains a central problem for the GPCR field. Contributing to this problem is the fact that most of the existing X-ray GPCR structures are of antagonist-bound receptors locked in inactive conformations. For the few GPCRs that have been captured in both inactive and active conformations, common structural changes include inward movements of transmembrane (TM) helices at the orthosteric pocket, rearrangements of common core residues through the helical bundle, and rigid-body outward movements of TM6 at the cytoplasmic surface to expose G protein binding epitopes (*Katritch et al., 2013*).

Beyond the static pictures of GPCR conformations seen in these crystal structures, experimental evidence for complex dynamic behavior has emerged from spectroscopic studies on purified receptors. Unlike rhodopsin, which exhibits an efficient and ordered photon-induced transition from the dark state conformation to metarhodopsin II (*Choe et al., 2011a*; *Choe et al., 2011b*), ligand-activated GPCRs generally show considerable basal activity that is reduced by inverse agonists. [19]F NMR and EPR experiments on the $β_2AR$ (*Liu et al., 2012a*; *Manglik et al., 2015*) and

$A_{2A}R$ (*Ye et al., 2016*) provided evidence for at least four different conformational states populated by these receptors, which are differentially stabilized by ligands of different efficacy. $^1H/^{13}C$ 2D NMR studies on $^{13}C\varepsilon$-methionine-labeled $\beta_2AR$ (*Nygaard et al., 2013*; *Kofuku et al., 2014*) showed that a high free-energy barrier exists between inactive and active conformations (with exchange on the millisecond timescale), and agonist binding alone is only capable of weakly populating the active conformation. These studies, in addition to similar measurements on μ-opioid receptor chemically modified to introduce NMR-active $^{13}C$-methyl labels onto lysine residues (*Sounier et al., 2015*), suggest that ligand-activated GPCRs are weakly coupled allosteric systems in which multiple energetic inputs (i.e. agonist and G protein binding) are required to predominantly populate the active conformation.

Within the slow millisecond-timescale global interconversion between functional GPCR conformations (*Vilardaga et al., 2003*), local structural rearrangements of sidechains in the receptor (referred to here as 'microswitches') have been observed in comparisons of inactive and active GPCR crystal structures (*Katritch et al., 2013*), NMR data (*Liu et al., 2012a*; *Manglik et al., 2015*; *Ye et al., 2016*; *Bokoch et al., 2010*) and MD simulations (*Vanni et al., 2009*; *Dror et al., 2011a*). The importance of repacking of sidechains from TM3, TM5, and TM6 beneath the ligand binding pocket, known as the 'conserved core triad,' was identified by comparison of structures of $\beta_2AR$ (*Rosenbaum et al., 2011*; *Rasmussen et al., 2011a*) and the μ-opioid receptor (*Huang et al., 2015*). Further down the TM bundle, an activated microswitch near the site of G protein binding has been found to represent a common feature of GPCRs that have been crystallized in both active and inactive conformations (*Venkatakrishnan et al., 2016*). A working hypothesis for GPCR activation is that a set of loosely coupled microswitches connecting the orthosteric pocket and G protein binding site are activated by agonists in a non-concerted fashion, and stabilizing subsets of these rearrangements can lead to alternate overall conformations that have different signaling properties.

One of the most powerful ways to probe sidechain dynamics and their contributions to biological processes is provided by NMR-based studies of isotopically-labeled methyl groups (*Ruschak and Kay, 2010*). To apply these techniques to study the sidechains in these microswitches, one approach would entail isoleucine/leucine/valine (ILV) labeling (*Goto et al., 1999*; *Gardner et al., 1998*) and perdeuteration of a wild-type GPCR, to generate proteins with $^1H/^{13}C$-labeled methyl groups within an otherwise $^2H/^{12}C$-labeled background. Such samples are ideally suited for acquiring $^1H/^{13}C$ methyl TROSY-based (*Tugarinov et al., 2003a*; *Ollerenshaw et al., 2003*) relaxation NMR data, allowing the quantitative determination of methyl order parameters and relative motion of sidechains on the ps-ns timescale (*Tugarinov et al., 2005*; *Ishima and Torchia, 2000*). Advances in labeling methodology and pulse sequences have enabled such measurements on large macromolecular systems, such as the 80 kDa enzyme malate synthase (*Tugarinov and Kay, 2003b*; *Sandala et al., 2007*), the 670 kDa archaeal proteasome core particle (*Sprangers and Kay, 2007*; *Religa et al., 2010*), and the 1 MDa GroEL/GroES complex (*Fiaux et al., 2002*; *Horst et al., 2005*). NMR spectroscopic interrogation of methyl groups has also been used to dissect the energetic contribution of sidechain entropy to important biological phenomena such as protein-protein interactions (*Marlow et al., 2010*) and transcription factor binding to double-stranded DNA (*Tzeng and Kalodimos, 2012*). These previous applications suggest that characterization of sidechain dynamics using methyl NMR approaches should be feasible for a GPCR purified in a detergent micelle, with an aggregate molecular weight in the 100–150 kDa range.

A major hurdle in carrying out such experiments is that perdeuteration and ILV labeling conventionally rely on expression in *E. coli* (*Goto et al., 1999*), while wild-type GPCRs typically require a eukaryotic host to ensure proper folding, glycosylation, trafficking, and stability in the plasma membrane. GPCR NMR studies have primarily used supplementation with labeled amino acids (*Nygaard et al., 2013*; *Kofuku et al., 2014*; *Okude et al., 2015*; *Isogai et al., 2016*; *Kofuku et al., 2012*) or covalent modification (*Liu et al., 2012a*; *Sounier et al., 2015*; *Bokoch et al., 2010*; *Eddy et al., 2016*) to incorporate isotopic probes into protein from Sf9 cells, and perdeuteration with incorporation of labeled ILV methyl probes is largely intractable for this established eukaryotic system. Another limitation is that purified wild-type GPCRs are typically prone to aggregation and are not highly thermostable, limiting acquisition times and sample concentration. While high quality 2D NMR spectra of GPCRs have been obtained for receptors with thermostabilizing mutations (*Isogai et al., 2016*), such mutations can affect the receptor's dynamics and functional properties, including basal activity and maximal G protein stimulation.

We have addressed many of these practical issues with our generation (*Clark et al., 2015*) of highly-deuterated $^1H/^{13}C$ Ile δ1-methyl labeled proteins in *Pichia pastoris*, a methylotrophic yeast (*Cereghino and Cregg, 2000*) that is adaptable to $D_2O$-based minimal media, will utilize the Ile precursor α-ketobutyrate, and enables expression of GPCRs at levels sufficient for crystallography (*Shimamura et al., 2011*; *Hino et al., 2012*; *Yurugi-Kobayashi et al., 2009*). Here we apply this labeling method to the wild-type human $A_{2A}R$, an important regulator of vascular function with a well-developed pharmacology including the natural hormone adenosine and high-affinity synthetic agonists and antagonists (*Preti et al., 2015*; *de Lera Ruiz et al., 2014*). $A_{2A}R$ has been crystallized in inactive (*Jaakola et al., 2008*; *Liu et al., 2012b*), intermediate (*Lebon et al., 2011*; *Xu et al., 2011*) and active (*Carpenter et al., 2016*) conformations, and has been studied by $^{19}F$ NMR to characterize its conformational equilibria (*Ye et al., 2016*). We expressed and purified milligram quantities of labeled wild-type human $A_{2A}R$ and were able to resolve 20 out of 29 expected peaks in the Ile δ1 region in $^1H/^{13}C$ TROSY HMQC experiments. Through structure-guided mutagenesis, we assigned four of these signals to specific functionally important residues, including I92$^{3.40}$ of the conserved core triad (*Huang et al., 2015*) and I292 at the cytoplasmic surface. We collected spectra in the presence of the high-affinity full agonist NECA and the inverse agonist ZM241385, and also measured the effects of monovalent cations ($Na^+$ versus $K^+$) on these spectra. We further carried out a modified triple-quantum (3Q) relaxation experiment (*Sun et al., 2011*) to quantify the relative fast ps-ns motions of these sidechains while the receptor is bound to ligands of opposing efficacy. These data represent an important first step towards understanding how agonists can activate the fast motions of specific sidechains to facilitate conformational changes of a GPCR.

## Results

The first challenge in collecting methyl NMR spectra on a wild-type GPCR was achieving high-level expression and milligram scale purification from *Pichia pastoris*. We initially cloned several GPCRs (including $A_{2A}R$) into a modified methanol-inducible expression vector with the wild-type MFα signal sequence at the N-terminus to direct incorporation into the membrane. Several receptors, such as the wild-type M2 muscarinic receptor and $A_{2A}R$, showed reliable and reproducible expression, however much of the expressed proteins were present in the immature unprocessed form (without MFα cleavage) indicating that they had not been properly localized to the plasma membrane (*Figure 1—figure supplement 1*). To improve the efficiency of production of mature, folded receptors, we made several modifications to the MFα sequence, based on biochemical and genetic precedents (*Rakestraw et al., 2009*; *Lin-Cereghino et al., 2013*) for improved processing efficiency (*Figure 1A*, Materials and methods). To validate that *Pichia*-expressed $A_{2A}R$ is functional, we carried out radioligand binding assays measuring [$^3H$]-ZM241385 saturation and NECA competition on membranes after cell wall disruption and zymolyase treatment. The $K_d$ for [$^3H$]-ZM241385 was 0.42 nM in NaCl-containing buffer, while the $K_i$ for NECA was 210 nM in low ionic strength and 420 nM in KCl-containing buffer, in agreement with previous binding studies (*Xu et al., 2011*; *Carpenter et al., 2016*; *Bertheleme et al., 2013*) on heterologously expressed $A_{2A}R$ (*Figure 1—figure supplement 2*).

We developed a purification protocol for $A_{2A}R$ from *Pichia*-derived membranes, in which we can solubilize and purify ~0.5 mg of monodisperse biochemically pure wild-type $A_{2A}R$ in dodecylmaltoside (DDM) detergent from 1 L of culture (*Figure 1—figure supplement 1*). Coupling this protocol with our previously-described labeling method (*Clark et al., 2015*), we were able to generate samples of wild-type $A_{2A}R$ that are highly deuterated both within the protein ($^2H$ at non-exchangeable sites except Ile δ1 methyl groups) and the surrounding buffer (99% $D_2O$ and deuterated DDM). We prioritized the use of a wild-type $A_{2A}R$ construct to report on the dynamics of a physiologically and functionally relevant GPCR sample. To confirm that the high level of deuteration did not alter the receptor's functional properties, we reconstituted the perdeuterated purified $A_{2A}R$ together with purified $G_s$ heterotrimer in phospholipid vesicles and carried out activation assays by [$^{35}S$]GTPγS binding. Saturating the receptor with the full agonist NECA led to an 8-fold stimulation of [$^{35}S$]GTPγS binding, and the inverse agonist ZM241385 produced a 50% reduction in basal activity, similar to values measured for protonated $A_{2A}R$ (*Figure 1B*). These data show that the perdeuterated $A_{2A}R$ is capable of activating $G_s$ at wild-type levels in vitro, and can serve as a valid model for GPCR dynamics.

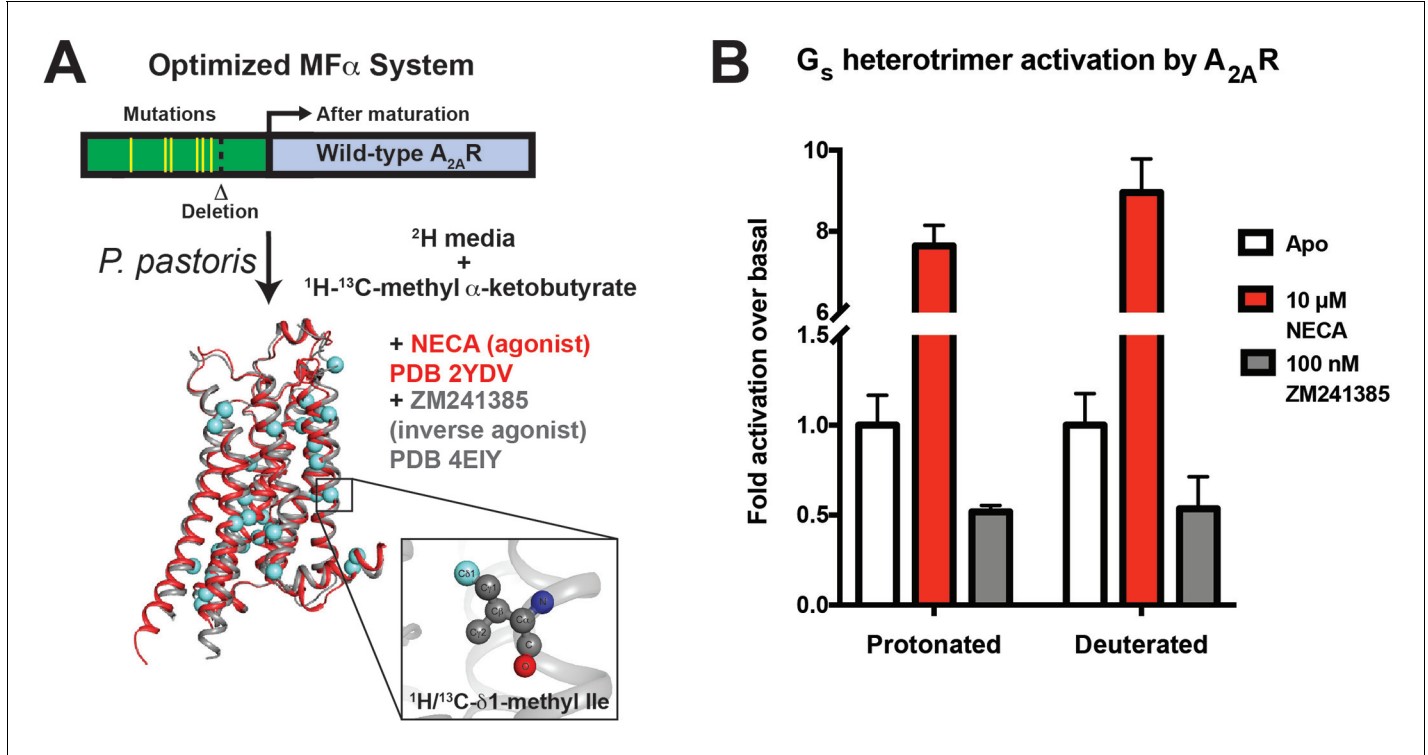

**Figure 1.** $A_{2A}R$ expression and $G_s$ activation. (**A**) Wild-type $A_{2A}R$ was expressed in *P. pastoris* as a fusion with an optimized version of the α-mating factor (MFα) signal sequence that contains mutations and a deletion that increase receptor trafficking to the plasma membrane. For NMR experiments, *Pichia* cultures were grown in deuterated media supplemented with ($^{1}$H/$^{13}$C-methyl) α-ketobutyrate to facilitate $^{1}$H/$^{13}$C labeling of Ile δ1 methyl groups in a deuterated background as previously described (**Clark et al., 2015**). Crystal structures of $A_{2A}R$ are shown complexed with agonist NECA (red, 2YDV (**Lebon et al., 2011**)) and inverse agonist ZM241385 (gray, 4EIY (**Liu et al., 2012b**), with isoleucine residues displayed as spheres. Labeled Ile δ1 carbon atoms are shown in cyan (see inset). (**B**) [$^{35}$S]GTPγS binding to purified, protonated or perdeuterated $A_{2A}R$ reconstituted with purified $G_s$ heterotrimer. Agonist NECA stimulates [$^{35}$S]GTPγS binding, while inverse agonist ZM241385 inhibits basal levels of binding.

DOI: https://doi.org/10.7554/eLife.28505.003

The following figure supplements are available for figure 1:

**Figure supplement 1.** $A_{2A}R$ processing and purification.

DOI: https://doi.org/10.7554/eLife.28505.004

**Figure supplement 2.** Ligand binding assays on $A_{2A}R$ yeast membranes.

DOI: https://doi.org/10.7554/eLife.28505.005

Our first NMR characterization of $A_{2A}R$ used $^{1}$H/$^{13}$C TROSY HMQC spectra of the protein in the presence of different ligands or $G_s$. We initially expressed and solubilized the receptor with the low-affinity antagonist theophylline, and exchanged other ligands onto the receptor during washing in the $Co^{2+}$-affinity and gel filtration steps (Materials and methods). In this way we were able to purify $A_{2A}R$ samples with the inverse agonist ZM241385, the full agonist NECA, or no ligand. The resulting 2D HMQC spectra, requiring ~100 μM GPCR and a minimum of 2 hr acquisition at 30°C on a cryo-probe-equipped 800 MHz spectrometer, are shown in **Figure 2**. Overall, in a liganded sample, we could resolve 20 peaks in the Ile δ1 region of the spectrum, out of 29 Ile residues present in our construct (28 from the receptor and one from the C-terminal Protein C tag). The chemical shifts of the agonist-bound and inverse agonist-bound $A_{2A}R$ are similar (**Figure 2A,B**), showing subtle changes that are comparable in magnitude to those observed previously for $^{13}$C-methyl-methionine probes in $β_2AR$ (**Kofuku et al., 2014**; **Kofuku et al., 2012**). The spectrum of the unliganded (apo) $A_{2A}R$ shows a loss of most of the well-dispersed peaks present in the ligand-bound spectra (**Figure 2C**), which could result from protein instability or conformational exchange on the μs-ms timescale. Lastly, we formed and isolated a high-affinity complex between purified NECA-bound $A_{2A}R$ and purified $G_s$ heterotrimer in the absence of GDP or GTP (**Figure 2—figure supplement 1**, Materials and methods). Like the apo spectrum, the G protein complex spectrum had fewer well-dispersed Ile peaks

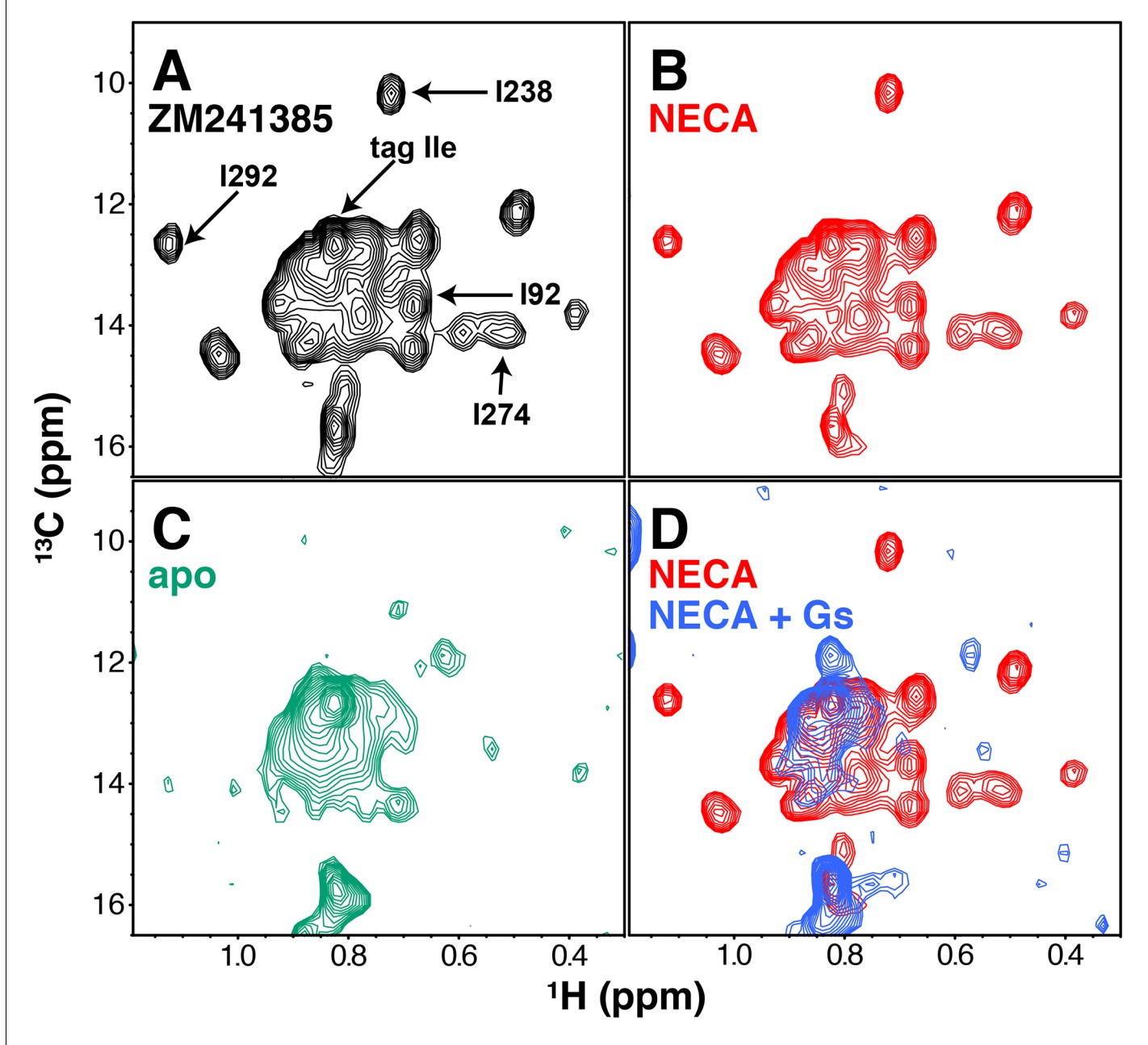

**Figure 2.** NMR spectra of A$_{2A}$R in different liganded states with NaCl. (A) $^1$H/$^{13}$C HMQC spectrum of Ile δ1-labeled WT A$_{2A}$R in DDM micelles and 150 mM NaCl with inverse agonist ZM241385 (black). Resonances assigned in this work (I92$^{3.40}$, I238$^{6.40}$, I274$^{7.39}$, I292, and the isoleucine residue in the protein C tag) are indicated with arrows. (B) $^1$H/$^{13}$C HMQC spectrum of Ile δ1-labeled WT A$_{2A}$R in DDM micelles and 150 mM NaCl with agonist NECA (red). See *Figure 2—figure supplement 3* for comparison of initial NECA spectrum with spectrum after 7.5 hr acquisition. (C) $^1$H/$^{13}$C HMQC spectrum of Ile δ1-labeled WT A$_{2A}$R in DDM micelles and 150 mM NaCl without ligand (green). (D) $^1$H/$^{13}$C HMQC spectra of Ile δ1-labeled WT A$_{2A}$R in DDM micelles and 150 mM NaCl with agonist NECA (red) and with agonist NECA and G$_s$ heterotrimer (blue).

DOI: https://doi.org/10.7554/eLife.28505.006

The following figure supplements are available for figure 2:

**Figure supplement 1.** A$_{2A}$R and G$_s$ complex formation.
DOI: https://doi.org/10.7554/eLife.28505.007

**Figure supplement 2.** Assignments of select Ile-δ1 methyl resonances in A$_{2A}$R.
DOI: https://doi.org/10.7554/eLife.28505.008

**Figure supplement 3.** Lifetime of A$_{2A}$R NMR sample.
DOI: https://doi.org/10.7554/eLife.28505.009

than receptor/ligand complexes (*Figure 2D*), consistent with the increase in molecular weight of the complex. We also note the appearance of a few distinct peaks (most prominently at 0.85 ppm $^1$H, 11.8 ppm $^{13}$C) in the G protein complex, consistent with the conformation of the complexed receptor being distinguishable from the apo receptor (*Rasmussen et al., 2011b*; *Carpenter et al., 2016*).

Assignment of $^1$H/$^{13}$C peaks in the Ile δ1 region by NMR was hampered by the modest dispersion of our spectra and the limited stability of A$_{2A}$R during data acquisition. Therefore, we took a site-specific mutagenesis approach, wherein we collected HMQC spectra of samples containing point mutations at a small subset of functionally important Ile sites (*Figure 2* and *Figure 2—figure supplement 2*). This process was complicated by the sensitivity of the receptor to mutations at several of these Iles: changes to Val, Leu, or Met often drastically reduced purification yields and/or led to significantly altered spectra; for an example, see the I106V spectra in *Figure 2—figure supplement 2*. Three of the assignments (I92$^{3.40}$, I238$^{6.40}$, and I292) could be made primarily through observation of ZM241385-bound mutant spectra. By comparing WT and I92V spectra in the presence of ZM241385, we see a clear loss of signal at ~0.68ppm/13.6ppm ($^1$H/$^{13}$C) that we can assign to the δ1 methyl of Ile92. The NECA-bound I92V spectrum is missing a subset of peaks compared to the ZM241385-bound spectrum, possibly due to compromised affinity of NECA for the I92V mutant (*Figure 1—figure supplement 2D*). Ile238 and Ile292 are well-dispersed peaks, allowing for unambiguous assignment through loss of peaks at 0.72 ppm/10.2 ppm and 1.12 ppm/12.7 ppm, respectively ($^1$H/$^{13}$C). The mutant spectra of I238$^{6.40}$ and I292 have reciprocal effects on each other, where I238V causes a full peak-width shift in the I292 signal (upfield in $^1$H), and I292L causes a ~ 0.5 peak-width shift in the I238 signal (upfield in $^{13}$C). The reciprocity between these mutant spectra is consistent with the proximity of these residues in the A$_{2A}$R structure. Mutation of I274$^{7.39}$ in the ligand binding pocket (within van der Waals contact of the ligands in the respective crystal structures) to Val or Met led to a disappearance of some peaks, reminiscent of the apo spectrum. However, comparison of ZM241385-bound I274M and I274V spectra narrowed the choice of the assignment of I274$^{7.39}$ to one of two peaks, with the greater solvent accessibility as revealed by solvent PRE effect (see below and Figure 4) of one peak serving as the deciding factor of assignment at 0.55 ppm/14.1 ppm ($^1$H/$^{13}$C). As well as disappearance of many peaks, the spectrum of I274M A$_{2A}$R in the presence of ZM241385 displays several apparent peak doublings, suggesting that the mutant protein is undergoing slow conformational exchange, potentially related to the slow interconversion between different inactive states of the receptor (*Ye et al., 2016*). In addition, we assigned a peak to the Ile residue of the C-terminal Protein C tag in our construct by comparing spectra between differently tagged constructs (not shown). The remaining unassigned peaks are referred to as a-o in the rest of this work (*Figure 3A*). The chemical shift changes between ZM241385-bound and NECA-bound A$_{2A}$R are small overall (*Figure 2A,B*). In the A$_{2A}$R-G$_s$ complex spectrum (*Figure 2D*), the peak for I92$^{3.40}$ appears to shift and simultaneously drop in intensity, strongly suggesting that I92$^{3.40}$ is involved in conformational changes on several timescales while the receptor is bound to G$_s$.

A$_{2A}$R contains a well-characterized Na$^+$ binding site within the receptor core (*Liu et al., 2012b*), and Na$^+$ has been shown to act as a negative allosteric modulator of A$_{2A}$R activation (*Gao and Ijzerman, 2000*). Due to the presence of 150 mM NaCl in the experiments shown in *Figure 2*, one possible explanation for the relatively small chemical shift differences between NECA- and ZM241385-bound spectra (*Figure 3A,B*) is that high [Na$^+$] is suppressing structural changes in the agonist-bound receptor. To test this possibility, we acquired HMQC spectra for A$_{2A}$R samples with 150 mM KCl substituting for NaCl. The spectrum of ZM241385-bound receptor in KCl is very similar to that in the presence of NaCl (*Figure 3C* and *Figure 3—figure supplement 1B*), with the peak assigned to I92$^{3.40}$ shifting slightly upfield in the $^{13}$C dimension. The spectrum of NECA-bound A$_{2A}$R, on the other hand, is dramatically different in KCl and NaCl (*Figure 3D*, *Figure 3—figure supplement 1D*), with multiple peaks shifting and decreasing in intensity. However, the KCl samples were significantly less stable than those in NaCl, showing markedly decreased spectral quality after 1 (NECA-bound) or 2 (ZM241385-bound) hours of acquisition. Given the weak signal-to-noise and poor stability of the KCl samples, we were only able to perform NMR relaxation experiments (below) on A$_{2A}$R in the presence of NaCl.

In addition to ligand-induced chemical shift changes, we used our system to characterize the solvent accessibility of residues within the ZM241385-bound A$_{2A}$R (with NaCl) using solvent paramagnetic relaxation enhancements (PREs). To do so, we used the soluble paramagnetic Gd$^{3+}$-DTPA probe (*Petros et al., 1990*), which enhances the relaxation rates of protons in a distance dependent

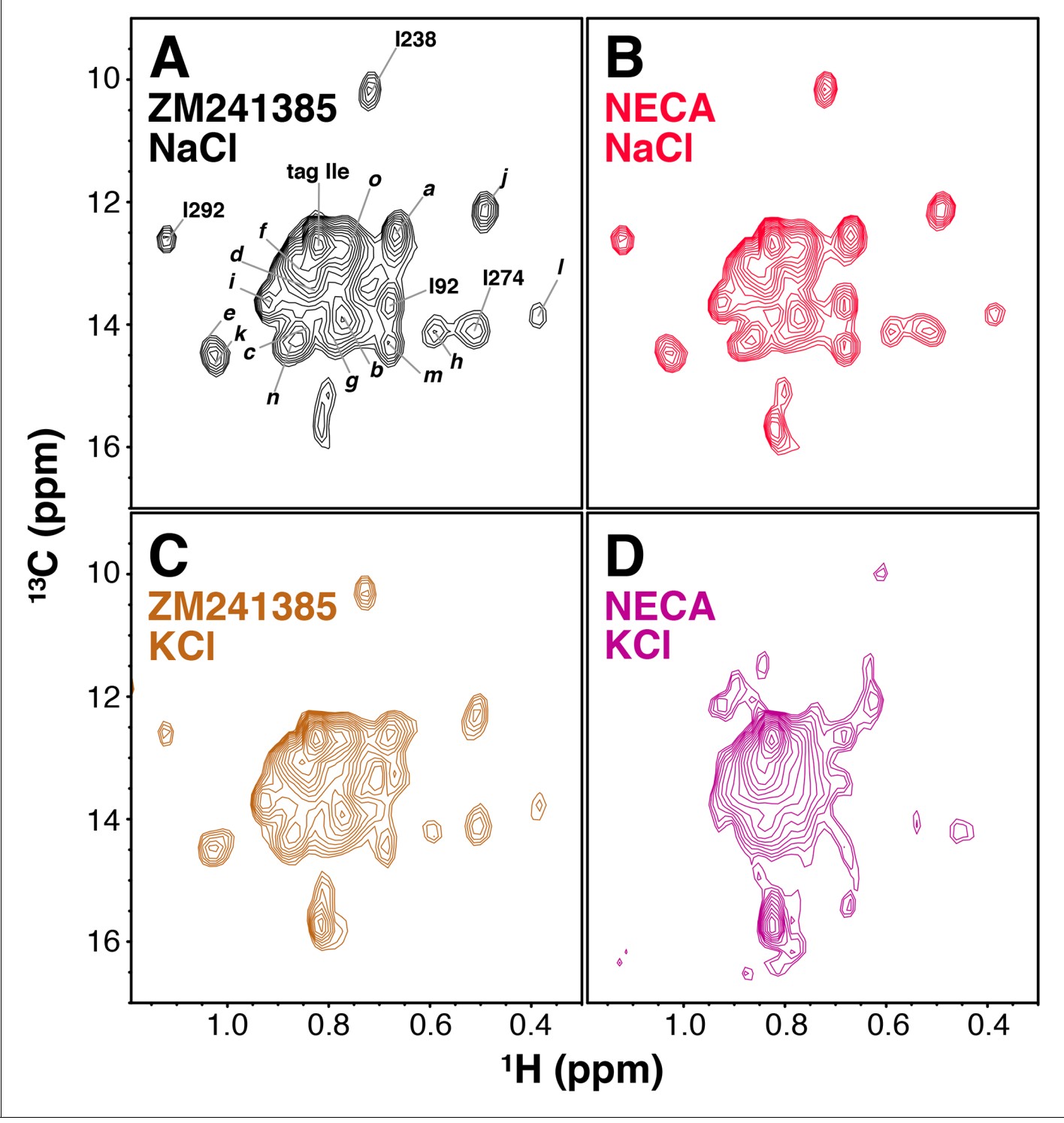

**Figure 3.** Ligand- and cation-dependent chemical shift changes in $A_{2A}R$. (A) $^{1}H/^{13}C$ HMQC spectrum of Ile $\delta1$-labeled WT $A_{2A}R$ in DDM micelles and 150 mM NaCl with inverse agonist ZM241385 (black). Residue numbers of assigned peaks and peak IDs of unassigned peaks are indicated in black. (B) $^{1}H/^{13}C$ HMQC spectrum of Ile $\delta1$-labeled WT $A_{2A}R$ in DDM micelles and 150 mM NaCl with agonist NECA (red). (C) $^{1}H/^{13}C$ HMQC spectrum of Ile $\delta1$-labeled WT $A_{2A}R$ in DDM micelles and 150 mM KCl with inverse agonist ZM241385 (brown). (D) $^{1}H/^{13}C$ HMQC spectrum of Ile $\delta1$-labeled WT $A_{2A}R$ in DDM micelles and 150 mM KCl with agonist NECA (purple).

DOI: https://doi.org/10.7554/eLife.28505.010

The following figure supplement is available for figure 3:

*Figure 3 continued*

**Figure supplement 1.** Overlays of ligand- and cation-dependent chemical shift changes in $A_{2A}R$.

DOI: https://doi.org/10.7554/eLife.28505.011

manner within roughly 15 Å of the protein/solvent interface (*Madl et al., 2011*). By acquiring $^1H/^{13}C$ TROSY HMQC spectra in the absence and presence of $Gd^{3+}$-DTPA, we could compare peak intensities between the two to establish solvent PRE levels. Examining our data, we see that the average peak intensity in the $Gd^{3+}$-DTPA containing spectrum is about 0.4 of that in the control spectrum, suggesting that the Ile δ1 methyl groups of $A_{2A}R$ in the DDM micelle are quite close to solvent on average (*Figure 4*). Of the four most strongly affected peaks, three peaks correspond to assigned residues (I238$^{6.40}$, I274$^{7.39}$, and I292). The accessibility of I274$^{7.39}$ is expected due this amino acid's presence in the solvent-exposed ligand-binding pocket. Broadening of I292 at the junction between TM7 and Helix 8 is also somewhat expected due to its position at the cytoplasmic surface. The broadening observed for I238$^{6.40}$ is more surprising, given that it is further embedded within the membrane and buried within the protein surface of $A_{2A}R$. Crystal structures of multiple GPCRs, including $A_{2A}R$, $β_2AR$, μOR, and rhodopsin have all revealed ordered solvent networks within the protein core in contact with the position equivalent to I238$^{6.40}$. In ZM241385-bound $A_{2A}R$, this residue is also in close proximity to the $Na^+$ site (*Liu et al., 2012b*), one packing layer toward the cytoplasmic surface. The solvent PRE effect seen for this residue could reflect breathing of the structure to expose this region to the $Gd^{3+}$-DTPA complex.

To characterize the motions of the Ile sidechains within $A_{2A}R$, we sought to carry out a triple quantum (3Q) relaxation experiment developed by Kay, Tugarinov and colleagues (*Sun et al., 2011*) to quantify sidechain dynamics in large macromolecules. In this experiment, two related series of 2D $^1H/^{13}C$ HMQC-based spectra were acquired, both of which start by generating transverse $^1H$ magnetization composed of single quantum (SQ) coherences. During a subsequent variable delay period,

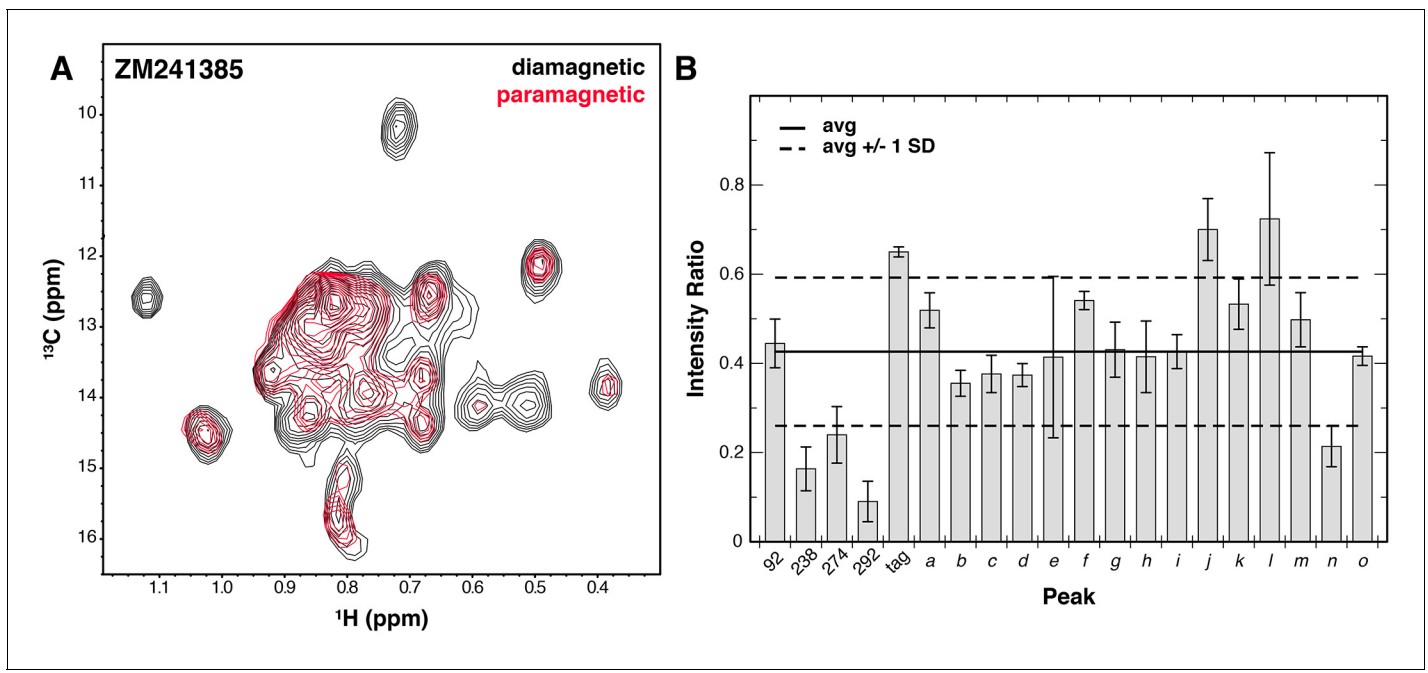

**Figure 4.** Solvent PRE analysis of $A_{2A}R$. (**A**) Solvent PRE $^1H/^{13}C$ HMQC spectra of Ile δ1-labeled WT $A_{2A}R$ in DDM micelles with inverse agonist ZM241385 in the presence (red) and absence (black) of paramagnetic $Gd^{3+}$-DTPA. Disappearance of peaks suggests solvent exposure. (**B**) Plot of intensity ratios between paramagnetic and diamagnetic samples for 20 peaks in the Ile δ1 region of the HMQC for $A_{2A}R$ complexed with inverse agonist ZM241385. The average (± 1 standard deviation) intensity ratio is shown as solid (dashed) black lines. Error bars show errors propagated from the noise of the NMR spectra. Assigned peaks are on the left of the plot, unassigned on the right. Peak IDs correspond to those in *Figure 3A*.

DOI: https://doi.org/10.7554/eLife.28505.012

intra-methyl $^1$H$-^1$H dipolar cross-correlated relaxation mechanisms lead to a portion of these SQ coherences evolving into 3Q coherences; the rate of this conversion ($\eta$) is proportional to the $S^2_{axis}$ methyl order parameter. Practically, the degree of SQ->3Q evolution can be quantitated by examining paired 'forbidden' and 'allowed' spectra, which report on the fraction of SQ coherences which do and do not evolve into 3Q coherences, respectively. As previously reported (*Sun et al., 2011*), this allows the simple calculation of the $\eta$ rate for each specific methyl group from the ratio of peak intensities in the forbidden and allowed spectra. Our initial attempts to apply this method to purified A$_{2A}$R were hampered by the fact that the wild-type receptor is insufficiently stable at 30°C to permit the measurement of the requisite paired spectra at many relaxation delays and with a high enough signal-to-noise ratio. We therefore explored the possibility of reducing the number of forbidden spectra required to accurately measure $\eta$ values. Using similarly $^2$H,$^{12}$C ($^1$H,$^{13}$C $\delta$1 methyl) labeled maltose binding protein (MBP) as a test case (Materials and methods), we collected 3Q datasets with different numbers of forbidden spectra paired with a constant number of allowed spectra (*Figure 5*, *Figure 5—figure supplement 1*). We found that pairing five allowed spectra with a single forbidden spectrum, we were able to faithfully recapitulate the $\eta$ values measured using five pairs of forbidden and allowed experiments (*Figure 5B,C*). This $\eta$ value is proportional to the $S^2_{axis}$ order parameter, which quantifies the amplitude of motion of the methyl group on the ps-ns timescale that is faster than global molecular tumbling.

With this modified analysis of 3Q relaxation data and our Ile $\delta$1 $^1$H/$^{13}$C-methyl-labeled and deuterated GPCR samples, we measured $\eta$ values for Ile sidechains in A$_{2A}$R bound to either ZM241385 or NECA in NaCl. The resulting values for the 20 Ile peaks and relative changes between the samples with different ligands are shown in *Figure 6*. The error in these measurements is large due to the limited signal-to-noise of our A$_{2A}$R spectra, and we did not attempt to convert these $\eta$ measurements to $S^2_{axis}$ values. However, in comparing the two datasets some features of the Ile dynamics can be discerned. The average $\eta$ value for the ZM241385-bound sample is higher than for NECA, indicative of greater overall rigidity of sidechains with inverse agonist. At the individual residue level, we observe a diversity of ligand-dependent changes in $\eta$, with some Ile residues becoming more rigid with agonist while most become more flexible. As an internal control, the peak we assigned to the C-terminal Protein C tag shows the lowest $\eta$ value measured (i.e. greatest flexibility) with little ligand-dependence. Among the other assigned peaks, I292 shows the largest difference between ligands, becoming more flexible in the NECA-bound sample. I92$^{3.40}$ and I274$^{7.39}$ display more modest increases in flexibility with agonist, while the dynamics of I238$^{6.40}$ are largely unchanged by ligand (*Figure 6B*). Interestingly, these differences in fast timescale dynamics occur in the presence of NaCl, in which chemical shifts for the agonist- and inverse agonist-bound states were nearly identical. To our knowledge, these data represent the first reported effort to experimentally quantify site-specific sidechain dynamics in a GPCR or comparable human integral membrane protein.

While we were able to measure sidechain dynamics in this challenging A$_{2A}$R sample, the associated errors are quite large. As such, we turned to molecular dynamics (MD) simulations to provide independent validation of our dynamics measurements. We extracted $S^2_{axis}$ order parameters for Ile $\delta$1 methyl groups from ~80 ns trajectories (post-equilibrium) of ZM241385- and NECA-bound A$_{2A}$R in DDM micelles, using overlapping 30-ns windows to estimate the standard deviation in the order parameters (*Figure 6—figure supplement 1*). The $S^2_{axis}$ order parameters for some methyl groups (for example, I60 and I80 in ZM241385; I106 and I302 in NECA) show significant variations across different windows, suggesting that slower timescale motions involving these residues contribute to the sidechain dynamics measured on the fast timescale. Interestingly, the differences between ZM241385- and NECA-bound dynamics at three of the four assigned sites (I238$^{6.40}$, I274$^{7.39}$, and I292) are qualitatively similar to what we see by NMR. The fourth site, I92$^{3.40}$, shows the opposite trend, with higher $S^2_{axis}$, i.e. more rigidity, in the NECA-bound state. The simulations were set up with bulk Na$^+$ ions, but none specifically occupying the binding site near I92$^{3.40}$, which could account for this discrepancy.

## Discussion

Sidechain dynamics represent an important functional component of protein behavior. In cases of protein-ligand (*Marlow et al., 2010*; *Frederick et al., 2007*) and protein-DNA (*Tzeng and*

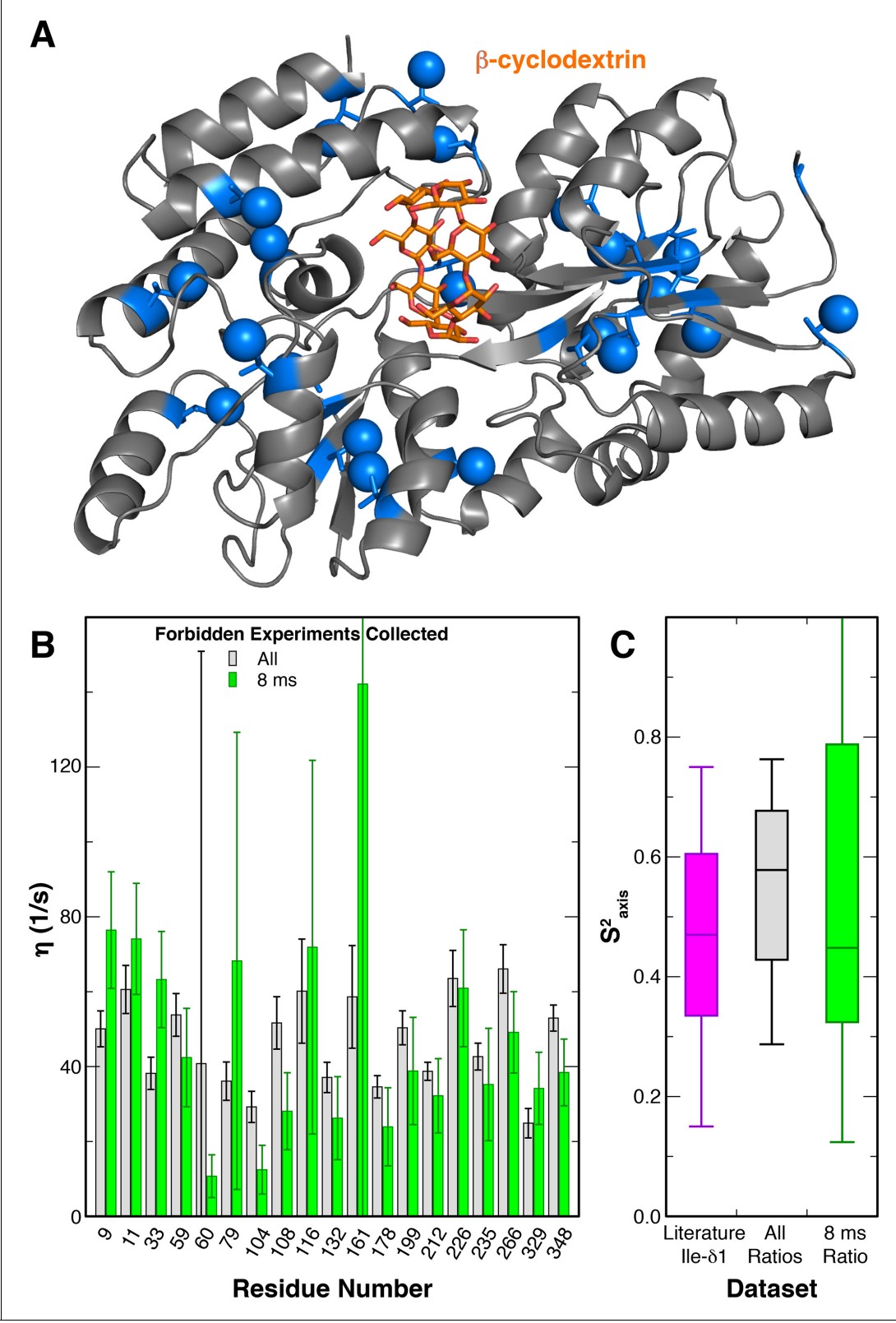

**Figure 5.** Modified 3Q relaxation for MBP. (**A**) Crystal structure of MBP complexed with β-cyclodextrin (PDB 1DMB[*Sharff et al., 1993*]). The β-cyclodextrin ligand is shown as orange sticks, while protein isoleucine residues are shown as blue sticks, with δ1 carbon atoms as spheres. (**B**) Plot of η values for each peak in MBP obtained from forbidden:allowed ratios at all five time points (gray) and the single 8 ms time point (green). Values calculated using the ratio at the single 8 ms time point show good agreement with those calculated using ratios at all five time points. (**C**) Box-and-

*Figure 5 continued on next page*

Figure 5 continued

whisker plot showing Ile δ1 $S^2_{axis}$ values from a set of globular proteins (*Mittermaier et al., 1999*) (purple) and the $S^2_{axis}$ values calculated from the η values using ratios at all time points (gray) and the single 8 ms time point (green).

DOI: https://doi.org/10.7554/eLife.28505.013

The following figure supplement is available for figure 5:

**Figure supplement 1.** Adaptation of 3Q methyl relaxation experiment to $A_{2A}R$ samples.

DOI: https://doi.org/10.7554/eLife.28505.014

*Kalodimos, 2012*) interactions that have been intensively studied, changes in entropy arising from modified sidechain dynamics in complexes were found to substantially contribute toward the overall free energy of binding. For GPCRs, changes in sidechain dynamics may play an energetic role in binding to ligands and G proteins, however we have focused on allosteric mechanisms connecting these two functionally important binding sites. To experimentally assess the roles of sidechain dynamics in a wild-type GPCR, we set out to create a labeled sample that would be amenable to relaxation NMR methods. The spectra we obtained for the labeled Ile δ1 methyl groups of perdeuterated $A_{2A}R$ are comparable or superior to previously published NMR spectra on other wild-type GPCRs, however the limited dispersion and signal-to-noise contributed to significant error in the relaxation values derived from the data (*Figure 6A*).

The $Na^+$ site first seen in the high-resolution structure of $A_{2A}R$ bound to ZM241285 is conserved throughout most Class A GPCRs, and $Na^+$ exerts a negative allosteric effect on $A_{2A}R$ activation by bridging residues on TM3 and TM7 and stabilizing the inactive conformation (*Liu et al., 2012b*). This effect can be observed pharmacologically as a NaCl-dependent decrease in agonist affinity (*Carpenter et al., 2016*; *Gao and Ijzerman, 2000*). For $A_{2A}R$ expressed in *P. pastoris*, we observe a 9-fold decrease in NECA affinity when membranes are incubated in 150 mM NaCl buffer versus 150 mM KCl buffer (*Figure 1—figure supplement 2*), similar to previously reported values (*Carpenter et al., 2016*). Several of the Ile residues that we assigned (i.e. I92[3.40] and I238[6.40]) are in close proximity to the $Na^+$ binding site, which should make for sensitive probes of local structure in this part of the receptor core. Our 2D NMR spectra indicate a strong dependency on combined lack of $Na^+$ and binding of agonist to stabilize significant structural changes in the receptor core (*Figure 3*). This observation is also consistent with the active conformation seen in the crystal structure of the $A_{2A}R$/mini-$G_s$ complex (*Carpenter et al., 2016*), which results in collapse of the $Na^+$ site due to inward movements of TM3 and TM7. In contrast to our data, the crystal structure of a NECA-bound thermostabilized $A_{2A}R$ mutant construct (*Lebon et al., 2011*) showed significant structural changes relative to the inactive conformation even with a high $[Na^+]$ beyond its $EC_{50}$ (~50 mM) for negative allosteric modulation (*Gao and Ijzerman, 2000*). Unfortunately the weak signal and stability of our samples in KCl precluded measurement of 3Q relaxation dynamics under these conditions.

For the $Na^+$-bound $A_{2A}R$ samples, we can correlate our observations of ligand-dependent changes in the relaxation rate of specific sites to previous structural and biophysical studies of GPCRs. I274[7.39] at the ligand binding site makes direct contact with the adenine or adenine analogue rings of both NECA and ZM241385 in their respective crystal structures (*Jaakola et al., 2008*; *Liu et al., 2012b*; *Lebon et al., 2011*; *Carpenter et al., 2016*) (*Figure 7A*). In our relaxation datasets, I274[7.39] was more rigid in terms of its fast sidechain motions with inverse agonist bound, and more flexible with NECA occupying the binding pocket. NMR experiments previously showed different ligand-dependent conformations of the extracellular surface region of $\beta_2AR$ bound to agonists compared to inverse agonists (*Bokoch et al., 2010*). In addition, MD simulations on $\beta_2AR$ predicted greater mobility of agonists relative to inverse agonists in the orthosteric pocket of the inactive-state structure (i.e. not bound to $G_s$) (*Dror et al., 2011b*). Our data suggests that the fast dynamics of residues at the orthosteric pockets of Class A GPCRs may be correlated with the functional efficacy of the bound ligand. The residue I92[3.40] is one of the three residues of the conserved core triad in $A_{2A}R$, along with P189[5.50] and F242[6.44], which interact at a layer beneath the orthosteric pocket further toward the cytoplasmic surface (*Figure 7B*). Consistent with the rearrangements of this microswitch region between the inactive and active conformations of multiple GPCRs, we observe a lower η value for I92[3.40] in the NECA dataset versus ZM241385, indicating greater sidechain flexibility

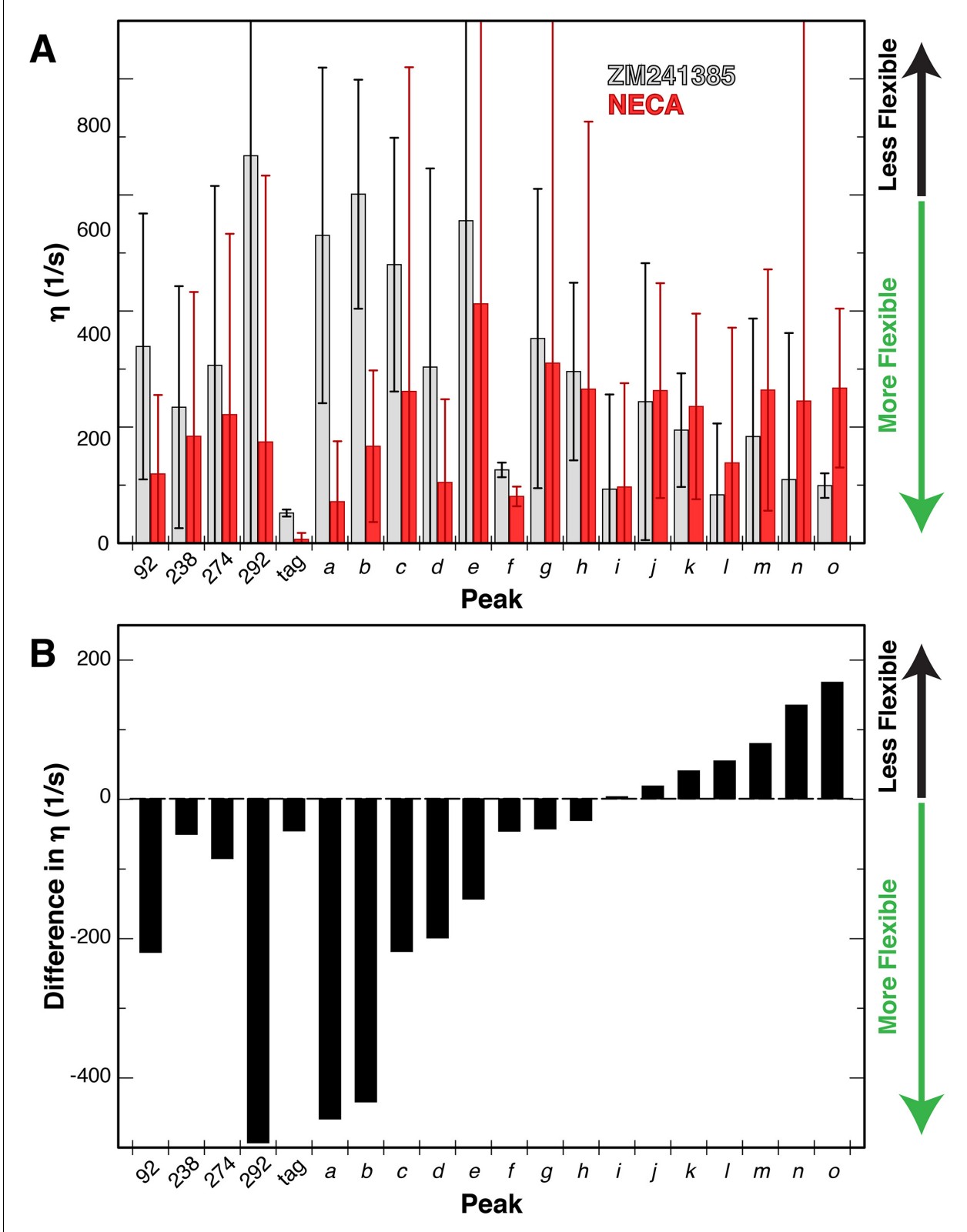

**Figure 6.** Changes in dynamics of methyl groups of $A_{2A}R$ in the presence of different ligands. (**A**) Plot of the $^1$H-$^1$H dipolar cross-correlation rate $\eta$, which is proportional to the $S^2_{axis}$ order parameter, for 20 peaks in the Ile $\delta1$ region for $A_{2A}R$ in DDM micelles with agonist NECA (red) and inverse agonist ZM241385 (grey). A higher $\eta$ value corresponds to more rigid methyl groups. Error bars show bootstrap errors of the fit parameters, propagated from the noise of the NMR spectra. Assigned peaks are on the left of the plot, unassigned on the right. Peak IDs correspond to those in

*Figure 6 continued on next page*

Figure 6 continued

Figure 3A. (B) Plot of the ligand-induced differences ($\Delta\eta = \eta[\text{NECA}] - \eta[\text{ZM241385}]$) for 20 peaks in the Ile $\delta$1 region for A$_{2A}$R in DDM micelles. Negative $\Delta\eta$ values, which are observed for the majority of peaks, indicate increased motions in the presence of agonist. Assigned peaks are on the left of the plot, unassigned on the right. Peak IDs correspond to those in *Figure 3A*.

DOI: https://doi.org/10.7554/eLife.28505.015

The following figure supplement is available for figure 6:

**Figure supplement 1.** Changes in dynamics of methyl groups of A$_{2A}$R in the presence of different ligands from MD trajectories.

DOI: https://doi.org/10.7554/eLife.28505.016

when A$_{2A}$R is agonist-bound. In several GPCRs, including β$_2$AR and μOR, the conserved core triad is unchanged at a static structural level with an agonist bound but without a G protein or nanobody to further stabilize the active conformation. In contrast, crystal structures of A$_{2A}$R bound to agonists alone (*Figure 7B*) showed an intermediate active-like conformation in the region surrounding I92$^{3.40}$ (*Lebon et al., 2011*; *Xu et al., 2011*). Our data indicate that low [Na$^+$] is required for NECA alone to stabilize structural rearrangements surrounding I92 (*Figure 3*, *Figure 3—figure supplement 1*). However agonist alone (even with high [Na$^+$]) is enough to at least promote the fast motions of the sidechains in this microswitch, which may reduce the activation energy for the observed packing rearrangement to the active conformation.

The residue I238$^{6.40}$ is situated further toward the G protein binding site at a critical region for GPCR activation, where it packs against TM7 and undergoes significant outward movement in the transition to the active conformation (*Figure 7C*). I238$^{6.40}$ is also one helical turn on TM6 above L235$^{6.37}$, which participates in a conserved microswitch between inactive and active conformations for multiple GPCRs (*Venkatakrishnan et al., 2016*). I292 is present at the linker between TM7 and Helix 8, where it packs against Y288$^{7.53}$ of the highly conserved NPXXY motif (*Katritch et al., 2013*) (*Figure 7D*). NMR studies of $^{13}$C-dimethyllysine-labeled μOR showed that peak broadening of a lysine probe in Helix 8 (near the position equivalent to I292) was more sensitive to agonist than probes at the cytoplasmic ends of TM5 and TM6 (21). Further, in the structure of A$_{2A}$R/mini-G$_s$ (*Carpenter et al., 2016*), the engineered G$_s$ protein makes direct contact with I292 at the TM7-Helix 8 junction. In our relaxation dataset, the η value of I238$^{6.40}$ is low (i.e. more flexible) and essentially independent of the ligand, while I292 undergoes a large change from more rigid to more flexible from ZM241385 to NECA. As mentioned above, the environment surrounding I238$^{6.40}$ is loosely packed in A$_{2A}$R structures, with an ordered solvent network that may allow for relative freedom of motion for this sidechain. In contrast, despite the solvent exposure of I292 (*Figure 4*), inverse agonist binding suppresses the fast dynamics of this residue relative to agonist binding (*Figure 6*). Outward movement of TM6 is one of the hallmarks of GPCR activation seen in crystal structures, and the cytoplasmic ends of TM6 and TM7 are separated from each other in the structures of β$_2$AR and A$_{2A}$R bound to G$_s$ (*Rasmussen et al., 2011b*; *Carpenter et al., 2016*). Our data suggest that the responsiveness of sidechain dynamics to ligands in these two regions may be largely uncoupled, and that the inverse agonist activity of ZM241385 could partly arise from its allosteric suppression of dynamics at the cytoplasmic end of TM7. Changes in η for other peaks in our dataset may provide further insights into the regulation of different regions of the receptor by ligands, depending on assignment of the other Ile residues in our spectra.

Beyond our studies of the A$_{2A}$R, we can now apply the methods for labeling and NMR spectroscopy described here to other GPCRs and eukaryotic membrane proteins. Since the microswitches discussed above were identified by comparison of different GPCR structures (*Huang et al., 2015*; *Venkatakrishnan et al., 2016*), it will be instructive to apply our methods to other receptors to see if the same patterns of ligand-dependent changes apply across the GPCR superfamily or change depending on the cognate ligand or preferred G protein signaling partner. Further, sidechain dynamics may be among the biophysical properties of GPCRs that are altered when receptors are bound to allosteric modulators or biased agonists, classes of ligands that are of increasing value and importance in GPCR pharmacology and drug development. In addition to GPCRs, many other disease-relevant human integral membrane proteins (such as ABC transporters and ion channels) are currently impossible to study by NMR methods, largely due to the challenges of expression, labeling, and perdeuteration in *E. coli*. Our approach has the potential to make these systems tractable for similar NMR measurements of sidechain dynamics.

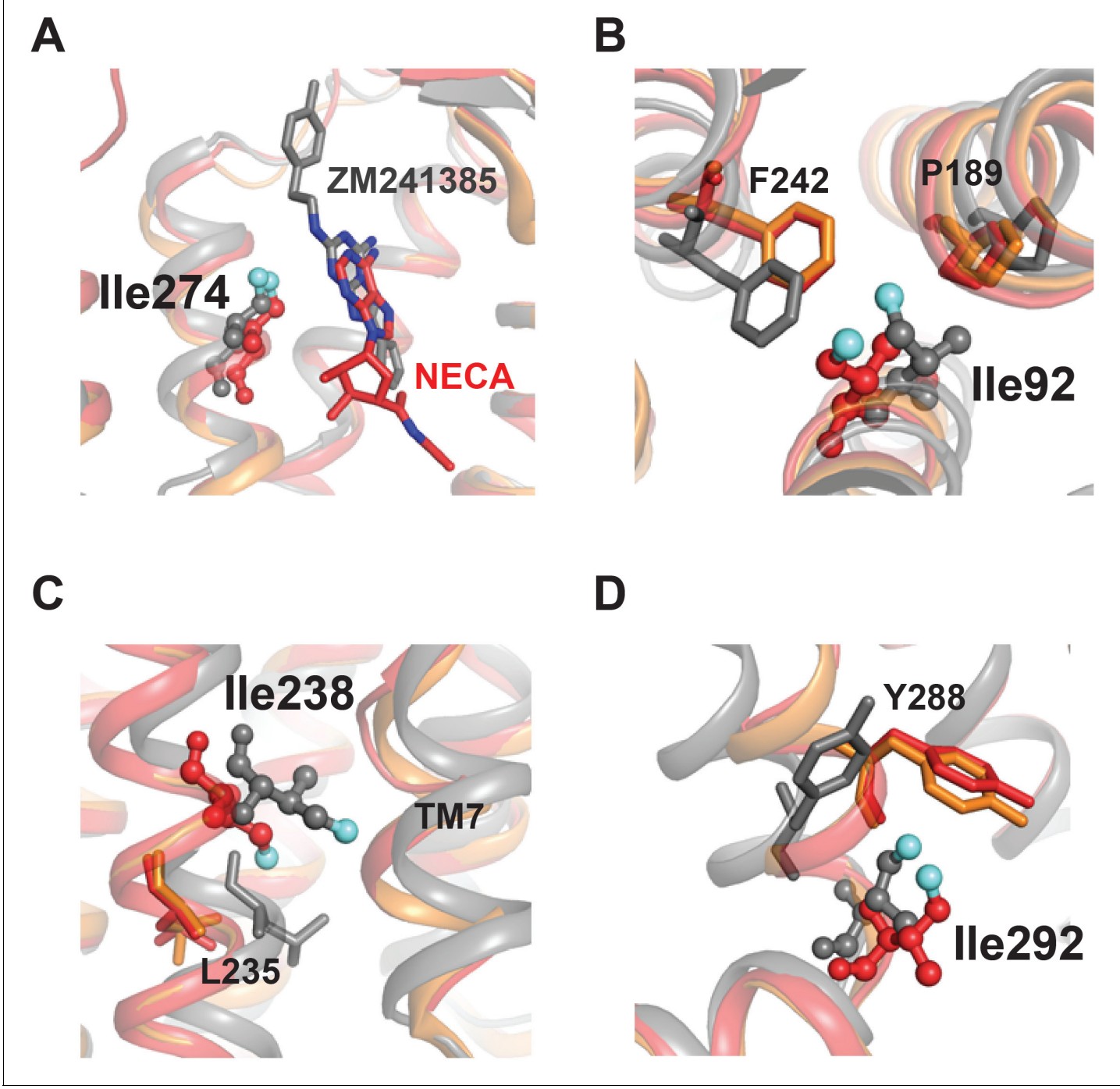

**Figure 7.** Structural contexts of isoleucine peak assignments. A$_{2A}$R ribbon diagrams are shown for structures solved in complex with inverse agonist ZM241385 (gray; PDB 4EIY[*Liu et al., 2012b*]), agonist NECA (red; PDB 2YDV[*Lebon et al., 2011*]), and agonist UK432097 (orange; 3QAK[*Xu et al., 2011*]). Assigned isoleucine residues are shown as spheres for the ZM241385- and NECA-bound structures using the same coloring scheme as in *Figure 1A*. Additional highlighted residues are shown as sticks in B-D. (**A**) Isoleucine 274 is part of the orthosteric binding pocket and makes direct contact with the adenine analogue and adenine rings in ligands ZM241385 and NECA, respectively (shown as sticks). (**B**) Isoleucine 92 sits below the binding pocket and interacts with nearby residues Pro189 and Phe242. Ile92, Pro189, and Phe242 all make structural rearrangements upon receptor activation. (**C**) On TM6, assigned peak Ile238 is one helical turn above Leu235. Leu235 is part of a conserved microswitch region that that undergoes outward motion during the transition to the active state. (**D**) Isoleucine 292 is at the cytoplasmic end of TM7 where it interacts with Tyr288, also part of the conserved NPXXY motif.

DOI: https://doi.org/10.7554/eLife.28505.017

## Materials and methods

### Construct design

The cDNA for wild-type human ADORA2A adenosine $A_{2A}$ receptor was cloned into the pPICZ vector for expression in *Pichia pastoris* with a modified MFα secretion signal combining previous precedents (*Rakestraw et al., 2009*; *Lin-Cereghino et al., 2013*). Modifications to the MFα greatly increased the amount of fully-processed receptor present at the plasma membrane. Briefly, the mutations V22A, G40D, L42S, V50A, V52A, and F55L were introduced, and residues 57–70 were removed from the signal sequence. The receptor expression construct was terminated at residue 316 and the N-glycosylation site at Asn154 was mutated to glutamine (*Jaakola et al., 2008*). The gene for the receptor was followed by a 8x His tag and a Protein C tag at the C-terminus. All point mutations were created using standard QuikChange protocols. Plasmids were linearized by incubation with *PmeI* (NEB) and inserted via electroporation into freshly prepared competent KM71H cells (Invitrogen, https://www.thermofisher.com/order/catalog/product/C18200). Clones were screened for integration efficiency with increasing amount of Zeocin (Invitrogen) in selection media and further selected through expression screens and western blots. The best expressing clones were stored as glycerol stocks at −80°C. As these yeast clones were only used for protein expression/purification and membrane preparation for binding assays, no authentication or mycoplasma contamination testing was performed.

### Expression of protonated cultures

For large scale growth in natural abundance media, a small amount of freshly streaked cells was inoculated into a 10 mL culture of BMGY media (1% glycerol, 100 mM potassium phosphate pH 6.0, 1.34% YNB (yeast nitrogen base), 0.004% histidine, $4 \times 10^{-5}$% biotin) and shaken at 28°C overnight at 250 rpm. The pilot culture was used to inoculate multiple liters of BMGY and shaken until saturation is reached ($OD_{600}$ ~20–30). The total culture was spun down in sterile bottles at 4000 rpm for 30 min and resuspended in equal volume of BMMY media (100 mM potassium phosphate pH 6.0, 1.34% YNB, 0.004% histidine, $4 \times 10^{-5}$% biotin) without methanol. Cultures continued shaking for ~8 hr at 28°C to allow for metabolism of residual glycerol. Protein expression was induced with the addition of 0.5% v/v methanol and the temperature was reduced to 20°C. An additional 0.5% methanol was added every 12 hr to maintain expression. Cells were harvested after 36–48 hr and pelleted by centrifugation. Pellets were stored at −80°C.

### Expression of deuterated cultures

For expression of cells in deuterated cultures, the cells were first adapted to deuterated media as follows. A small amount of freshly streaked cells was inoculated into a 50 mL culture of BMGY containing 90% $D_2O$/10% $H_2O$ (Cambridge Isotope Laboratories, Inc., Tewksbury MA). The culture was shaken at 28°C until an $OD_{600}$ of 8–10 was reached (typically ~24 hr). Once that OD was reached, 200 μL of the 90%/10% culture was inoculated into 50 mL of BMGY media made with 100% $D_2O$ and protonated glycerol. The culture was shaken at 28°C until an $OD_{600}$ of 8–10 was reached (typically ~48 hr). 200 μL of the 100% culture was inoculated into 50 mL of BMGY media made with 100% $D_2O$ including $d_8$-glycerol as the carbon source (Cambridge Isotope Laboratories, Inc.). This culture was incubated until reaching an $OD_{600}$ of ~10 and the entire culture was inoculated into large scale cultures of BMGY media again made with 100% $D_2O$ including $d_8$-glycerol. The large scale cultures were shaken until saturation ($OD_{600}$ of ~20–30) and then spun down in sterile bottles at 4000 rpm for 30 min. The cells were resuspended in BMMY made in 100% $D_2O$ without methanol and continued shaking for 12–16 hr to metabolize residual $d_8$-glycerol. One hour prior to induction, 200 mg/L of labeled α-ketobutyric acid (methyl-[13]C, 99%; 3,3-$D_2$, 98%; Cambridge Isotope Laboratories, Inc.) was added to the culture. Ten minutes prior to induction, dry theophylline was added to the culture to a final concentration of 4 mM. Protein expression was induced with the addition of 0.5% $d_4$-methanol (Cambridge Isotope Laboratories, Inc.) and the temperature was reduced to 20°C. Expression was maintained by further additions of 0.5% $d_4$-methanol every 12 hr, and cells were harvested by centrifugation after 36–48 hr and stored at −80°C.

## Purification

Cell pellets were thawed and resuspended in lysis buffer (PBS containing 10% glycerol, 4 mM theophylline, 2 mM EDTA, and protease inhibitors (160 µg/mL benzamidine, 2.5 µg/mL leupeptin, 1 mM PMSF, 1 µM E-64)). Cells were passed through a high-pressure microfluidizer (Microfluidics M-110P) three times at 24,000 psi with a cooling period between passes. LongLife Zymolyase (G-Bio Sciences) was added to the total lysate at a concentration of 15 U/mL and stirred at 37°C for 1 hr. Total membranes were isolated by centrifugation at 140,000 rcf for 30 min and then washed by douncing in an equal volume of lysis buffer followed by an additional centrifugation step. Membranes were then resuspended in hypotonic buffer (10 mM HEPES pH 7.5, 2 mM EDTA, 4 mM theophylline, protease inhibitors) by douncing and stirred at 4°C for 30 min, followed by centrifugation again to isolate membranes. Membranes were dounced in buffer containing 500 mM NaCl, 50 mM HEPES pH 7.5, 20% glycerol, 1% DDM (Anatrace), 4 mM theophylline, and protease inhibitors and stirred for 2 hr at 4°C. Insoluble material was spun out by centrifugation at 140,000 rcf for 30 min. The resultant supernatant was incubated with TALON resin (Clontech, Mountain View CA) pre-equilibrated in 250 mM NaCl, 50 mM HEPES 7.5, 0.05% protonated DDM, 5% glycerol, 4 mM theophylline, and 30 mM imidazole. Additional imidazole was added to the supernatant to the final concentration of 30 mM to minimize background binding. Batch binding continued overnight at 4°C.

Following batch binding, the resin was washed with a series of buffers to exchange the protonated DDM into deuterated DDM and exchange on the high affinity ligands ZM241385 or NECA. Buffers were made in $D_2O$ and all contain 250 mM NaCl, 50 mM HEPES pH 7.5, 5% glycerol, 0.05% DDM (protonated or deuterated; Anatrace), 20 mM imidazole, and 10 µM ZM241385 or 20 µM NECA (Tocris). Buffers contain different ratios of protonated:deuterated detergents, and were added sequentially: (A) 4:0; (B) 3:1; (C) 2:2; (D) 1:3; (E) 0:4. Protein was eluted from TALON with buffer E + 250 mM imidazole. Eluted $A_{2A}R$ was concentrated in 100 kDa MWCO Amicon concentrators (Millipore) and injected on a Superdex200 column (GE Healthcare, Chicago IL) equilibrated in 150 mM NaCl, 20 mM HEPES pH 7.5, 0.05% deuterated DDM, and 10 µM ZM241385 or 20 µM NECA made in $D_2O$. Samples for KCl experiments were purified in the same manner, only substituting 150 mM KCl instead of NaCl in the SEC buffer.

Previous studies have shown the importance of using cholesterol hemisuccinate (CHS) as a component of the micelle to preserve $A_{2A}R$ function (*Weiss and Grisshammer, 2002*) and increase the thermostability of the receptor (*Liu et al., 2012b*). However, when CHS was included in our purification, we observed significant artifacts in the Ileδ1 region of our NMR spectra that hampered data collection and analysis. Ye et al. utilized an XAC ligand affinity column as an additional chromatography step during purification (*Ye et al., 2016*) that we did not include in this study. Our purification is similar to what has been published in structural studies of $A_{2A}R$, following a general purification scheme of IMAC followed by size exclusion chromatography prior to crystallization in detergent (*Hino et al., 2012*; *Carpenter et al., 2016*; *Sun et al., 2017*) or lipidic cubic phase (*Jaakola et al., 2008*; *Liu et al., 2012b*; *Xu et al., 2011*). These structures contain $A_{2A}R$ bound to agonists, antagonists, or in complex with an engineered 'mini-Gs' protein. Given our biochemical evidence (*Figure 1—figure supplement 1*, *Figure 2—figure supplement 1*), we are confident that our purification protocol outlined above produces a high level of folded and functional $A_{2A}R$, which is able to efficiently bind to $G_S$ in solution.

## G protein expression and purification

$G_s$ heterotrimer was purified from *Trichoplusia ni* cells grown in ESF921 media (Expression Systems). Cells were lysed in hypotonic buffer containing 10 mM Tris pH 7.4, 100 µM $MgCl_2$, 5 mM β-mercaptoethanol, 10 µM GDP, and protease inhibitors. After centrifugation, membranes were dounced and solubilized in buffer containing 100 mM NaCl, 20 mM HEPES pH 7.5, 1% sodium cholate, 0.05% DDM, 5 mM MgCl2, 5 mM β-mercaptoethanol, 10 µM GDP, 5 mM imidazole, protease inhibitors, and a 1:150,000 vol dilution of CIP (NEB).

The soluble material was incubated with pre-equilibrated $Ni^{2+}$-NTA resin for 2 hr at 4°C. Detergent exchange into deuterated DDM was performed on-column by mixing volumes of buffer E1 (100 mM NaCl, 20 mM HEPES pH 7.5, 1% sodium cholate, 0.05% deuterated DDM, 5 mM $MgCl_2$, 5 mM β-mercaptoethanol, 10 µM GDP, 20 mM imidazole, protease inhibitors) and buffer E2 (50 mM NaCl, 20 mM HEPES pH 7.5, 0.1% deuterated DDM, 1 mM $MgCl_2$, 5 mM β-mercaptoethanol, 10 µM GDP,

20 mM imidazole, protease inhibitors) in the following E1:E2 ratio: 1:1; 1:3; 1:9; 1:19 at a flow rate of 1 mL/min. $G_s$ was eluted with buffer E2 supplemented with 200 mM imidazole. Following elution, $MnCl_2$ was added to a final concentration of 1 mM, and the pooled eluate (~10 mL volume) was incubated with 10 µL lambda phosphatase, 1 µL CIP, and 1 µL Antarctic phosphatase for 30' at 4°C. Sample was diluted with buffer E2 to reduce imidazole concentration and applied to a pre-equilibrated 2 mL Q-sepharose column. The resin was washed with 6 CV of 100 mM NaCl, 20 mM HEPES pH 7.5, 0.04% deuterated DDM, 1 mM $MgCl_2$, 100 µM TCEP, 10 µM GDP and eluted with wash buffer supplemented with an additional 250 mM NaCl. Eluted sample was concentrated in a 10 kDa MWCO concentrator to 1 mL and diluted 1:1 with buffer containing 20 mM HEPES pH 7.5, 1.5 mM EDTA, 1.65 mM $MgCl_2$, 100 µM TCEP, 1 µM GDP to reduce final DDM and NaCl concentrations. Glycerol was added to 20%, and aliquots were flash frozen in $LN_2$ and stored at −80°C until needed. Functional $G_s$ was conservatively estimated to constitute at least 40% of the total purified sample.

## $A_{2A}R$-$G_s$ complex formation

$A_{2A}R$-$G_s$ complex formation was carried out as follows. SEC-purified $A_{2A}R$ was mixed with $G_s$ at a 1:8 w/w ratio in the presence of 1 mM EDTA, 3 mM $MgCl_2$, 10 µM NECA, 1 mM $MnCl_2$, and a 1:100 dilution of lambda phosphatase (NEB). The complex was incubated at room temperature for 1 hr, then apyrase (NEB, Ipswich MA) was added at a 1:2000 vol dilution followed by an additional hour incubation at room temperature. $CaCl_2$ was added to a final concentration of 2 mM and loaded at a flow rate of 10 mL/hr on Protein C antibody resin (Sigma-Aldrich, St. Louis MO) pre-equilibrated in 150 mM NaCl, 20 mM HEPES pH 7.5, 0.05% DDM, 2 mM $CaCl_2$, 10 µM NECA. ProC resin was washed with 1 CV buffer at 10 mL/hr, followed by 5 CV buffer at 30 mL/hr. Complex was eluted in 2 CV with 150 mM NaCl, 20 mM HEPES pH 7.5, 0.05% DDM, 10 µM NECA, 5 mM EDTA, 0.2 mg/mL ProC peptide (EDQVDPRLIDGK). Freshly made TCEP was added to a final concentration of 100 µM and was spin-concentrated prior to injection on a Superdex200 column equilibrated in 150 mM NaCl, 20 mM HEPES pH 7.5, 0.05% DDM, and 10 µM NECA.

$A_{2A}R$-$G_s$ complex formation for NMR proceeded as above with a few modifications. $G_s$ heterotrimer purification was carried out as above with the final exchange steps containing deuterated DDM instead of protonated DDM. Deuterated, labeled, SEC-purified $A_{2A}R$ was mixed with $G_s$ at a 1:8 w/w ratio in the presence of 3 mM $MgCl_2$ in a final volume of 1 mL. The complex was incubated at room temperature for five minutes, and then a 1/2000 vol of apyrase (NEB) was added. The complex was incubated for an additional hour at room temperature, and then was injected on a Superdex200 column incubated in 150 mM NaCl, 20 mM HEPES pH 7.5, 0.05% deuterated DDM, 20 µM NECA. Peak fractions were concentrated to ~100 µL for NMR experiments.

## NMR spectroscopy and analysis

NMR spectra were collected at 30°C on 50–100 µM $A_{2A}R$ (complexed with ZM241285 or NECA) and 100 µM MBP (complexed with β-cyclodextrin) using a Bruker AVANCE III HD 800 MHz spectrometer with a cryogenically-cooled TCI probe. All NMR samples were approximately 100 µl in a 3 mm Shigemi tube. NMR data were processed using NMRpipe (*Delaglio et al., 1995*) and analyzed using NMRViewJ (*Johnson, 2004*). Parameters derived from relaxation data were obtained with a non-linear least-squares fitting programmed in the Python SciPy library (*Jones et al., 2001*; *More, 1977*). Errors in these parameters were generated using a nonparametric Monte Carlo bootstrap approach (*Efron, 1981*) with 1000 simulated datasets, each of which consisted of synthetic datapoints with generated values centered on the measured peak intensities (and errors determined from the noise of the experimental spectra). Each simulated dataset was individually fit; parameters from these fits were averaged to generate the reported values (as average ± standard deviation). Unassigned isoleucine methyl peaks in $A_{2A}R$ are referred to by a peak ID (*Figure 3A*), sorted by increasing difference between inverse agonist- and agonist-bound relaxation rates (*Figure 6B*).

$^1$H/$^{13}$C-HMQC spectra of the methyl region were collected with a $^{13}$C spectral width of 22 ppm (8 ppm) centered at 14 ppm (9.5 ppm) for $A_{2A}R$ (MBP), with 32 complex pairs collected. Due to short sample lifetimes, some $A_{2A}R$ HMQC spectra were collected as several sequential experiments with 64 scans per t1 point; if peak intensities and locations in these spectra remained consistent (for example, see *Figure 2—figure supplement 3*), they were summed after processing. Solvent PRE experiments were carried out by collecting an HMQC spectrum on an $A_{2A}R$ sample, then adding

Gd$^{3+}$ complexed with diethylenetriaminepentaacetic acid (DTPA; Sigma-Aldrich) to a final concentration of 1 mM and collecting a matched HMQC spectrum. The PRE effect was measured as the ratio of peak intensities in the paramagnetic sample to those in the diamagnetic sample.

Triple quantum (3Q) relaxation experiments were conducted using a pulse sequence kindly provided by Prof. Lewis Kay (University of Toronto) (*Sun et al., 2011*). The carrier was centered on the methyl region (0.8 ppm) in the $^1$H dimension, while $^{13}$C dimension center and spectral width were as above for HMQC spectra. The 3Q relaxation experiments were run as pseudo-4D experiments with the relaxation delays and alternating forbidden/allowed experiments as the third and fourth dimensions. An NMRpipe (*Delaglio et al., 1995*) script was used to divide the data and process each spectrum. For MBP, spectra were collected with relaxation delays of 0.8, 2, 4, 8, and 16 ms for both forbidden and allowed experiments. Peaks (corresponding to I317 and I333) with low intensities (defined as less than ten-fold greater than noise in the 16 ms forbidden spectrum) were not used for relaxation analysis. Peak intensities were measured using the NMRVIewJ (*Johnson, 2004*) Rate Analysis module by fitting each peak to an ellipse and calculating the volume. Ratios of peak intensities were fit to the following equation as a function of relaxation delay (*Sun et al., 2011*):

$$\left| \frac{I_{forb}}{I_{all}} \right| = \frac{3N_{all}}{4N_{forb}} \frac{\eta \tanh\left(\sqrt{\eta^2 + \delta^2} T\right)}{\sqrt{\eta^2 + \delta^2} - \delta \tanh\left(\sqrt{\eta^2 + \delta^2} T\right)} \tag{1}$$

where N is the number of scans for each experiment, T is the relaxation delay, $\delta$ (<0) is a term for the coupling between rapidly and slowly decaying single-quantum coherences, and $\eta$ is a relaxation rate, defined as the difference between slow and fast relaxation rates of single-quantum transitions for methyl protons:

$$\eta = \frac{R_{2,H}^F - R_{2,H}^S}{2} \propto \tau_c S_{axis}^2 \tag{2}$$

proportional to the methyl axis order parameter $S_{axis}^2$ and correlation time $\tau_c$. All relaxation rates were constrained to be >0 while $\delta$ was constrained to be <0.

Due to short lifetimes of A$_{2A}$R samples (~12–14 hr at 30°C), the relaxation experiment was modified and run with five allowed experiments (0.8, 2, 4, 8, and 14 ms) and fewer forbidden experiments. To extract values of $\eta$, the forbidden:allowed peak intensity ratios at those few relaxation delays were fit to *Equation 1* as above. Simultaneously, the allowed experiment peak intensities were fit to:

$$I_{all} = A \frac{3}{2} \left[ \exp\left(-R_{2,H}^S T\right) + \exp\left(-R_{2,H}^F T\right) \right] \tag{3}$$

where A is a scaling constant, T is the relaxation delay, and R$^S$ and R$^F$ are as above (*Figure 5—figure supplement 1*).

To determine which relaxation delays for the limited forbidden experiments gave values of $\eta$ that best agreed with those determined from a full dataset, the MBP relaxation dataset was analyzed using all five allowed experiments and different combinations of 1 or two forbidden experiments. The differences in $\eta$ values for each peak, as well as the sum of squared differences across all peaks, were compared for each analysis and a single relaxation delay of 8 ms was selected for A$_{2A}$R relaxation experiments (*Figure 5*, *Figure 5—figure supplement 1*). For processing, two separate scripts were used to extract the five allowed experiments and sum up the five forbidden experiments. Allowed peak intensity and forbidden:allowed peak intensity ratios were calculated and fit to *Equations 1 and 2*, respectively, as above, to derive values for $\eta$ at each peak.

## Computational methods
### Molecular constructs
To construct systems for atomistic molecular dynamics (MD) simulations, we used the X-ray structures of the human A$_{2A}$ GPCR in complex with agonist NECA or inverse agonist ZM241385 (PDB accession codes 2YDV (*Lebon et al., 2011*) and 4EIY (*Liu et al., 2012b*), respectively). The thermostabilizing agents present in these X-ray models were removed and thermostabilizing mutations in

the $A_{2A}R$ introduced during the crystallization experiments were reverted back to the corresponding wild type residues using VMD mutator plugin (*Humphrey et al., 1996*). Several short residue segments missing from the X-ray structures were added with Modeller (*Sali and Blundell, 1993*) to complete the full-length atomistic models of the human $A_{2A}R$. The $Na^+$ ion bound to the protein in the 4EIY structure was removed so that the effects of the agonist and the inverse agonist on the protein dynamics could be directly compared.

Using the protocol we have described earlier (*Khelashvili et al., 2013*), the NECA- and ZM241385-bound $A_{2A}R$ structures were inserted into a detergent micelle containing 246 DDM molecules. The resulting proteomicelles were placed in ~$157^3$ $Å^3$ size cubic water box containing 150 mM NaCl salt. The final molecular systems contained ~330,000 atoms and resulted in detergent concentration of ~0.1 M, well-above established critical micelle concentration (CMC) for DDM (170 µM) (*Kaufmann et al., 2006*).

## Force fields and MD simulations

The all-atom MD simulations were performed with the NAMD 2.7 package (*Phillips et al., 2005*) using the all-atom CHARMM27 force-field with CMAP corrections for proteins (*Brooks et al., 2009*), and a CHARMM-compatible force-field parameter set for detergents (*Abel et al., 2011*). CHARMM-suitable parameters for NECA and ZM241385 compounds were generated with the program MATCH (*Yesselman et al., 2012*). Molecular constructs were initially equilibrated using a two-phase protocol implemented by us earlier for MD simulations of leucine transporter (LeuT)/micelle complexes (*Khelashvili et al., 2013*; *LeVine et al., 2016*): (i) short energy minimization was carried out during which protein, water, and ion atoms were fixed and the coordinates of only DDM molecules were allowed to evolve freely; and (ii) 1.5 ns long MD simulations were conducted with the protein backbone harmonically constrained. The constraints were released gradually, in 0.5 ns steps, with decreasing force constants of 1, 0.5, and 0.01 kcal/ (mol·$Å^2$).

After the equilibration phase, both the NECA-bound and ZM241385-bound $A_{2A}R$ systems were subjected to ~80 ns unbiased MD simulations. Integration steps were 1 fs for the equilibration stage and 2 fs thereafter. The simulations implemented PME for electrostatics interactions (*Essmann et al., 1995*) and were carried out in NPT ensemble under isotropic pressure coupling conditions, and at 310 K temperature. The Nose-Hoover Langevin piston algorithm (*Phillips et al., 2005*) was used to control the target p=1 atm pressure with the LangevinPistonPeriod set to 100 fs and LangevinPistonDecay to 50 fs. The first ~25 ns of each trajectory was discarded based on the convergence of the Cα RMSD.

Each MD trajectory was split into six 32-ns windows overlapping by 28 ns. The $S^2_{axis}$ order parameters for Ile δ1 methyls were calculated from Cγ1-Cδ1 bond vectors extracted from four of these windows using the trjtool software (*Xue et al., 2014*; *Bremi et al., 1997*). The average ±standard deviation of these order parameters over the four windows is reported for each Ile δ1 methyl.

PDB files used in MD simulations and referenced throughout this manuscript (i.e. in *Figure 7*) can be accessed at http://www.rcsb.org.

## Membrane binding

Ligand binding experiments on membranes containing $A_{2A}$ receptor were carried out based on previously published protocols (*Xu et al., 2011*). *Pichia* cells expressing each construct were used to generate membranes as follows. Cells were resuspended in lysis buffer (PBS containing 10% glycerol, 2 mM EDTA, and protease inhibitors) and incubated for 2 hr at 37°C with Zymolyase 20T (AMS Bio, Abingdon UK) at a final concentration of 50 U/mL. Crude membranes were isolated through centrifugation at 40,000 rcf for 30 min and dounced in storage buffer (150 mM NaCl, 50 mM HEPES pH 7.5, 10% glycerol, protease inhibitors). Large cell debris was removed by a low speed spin at 1000 rcf for 10 min, and remaining membranes were subjected to a high speed spin at 140,000 rcf for 30 min. Pellets were dounced in a minimal volume of storage buffer, flash frozen in $LN_2$, and stored at −80°C until needed. Saturation binding was carried out by incubating 1.5–2 µg of membranes with different concentrations of [$^3$H]-ZM241385 (50 Ci/mmol, American Radiolabeled Chemicals, Inc., St. Louis MO) between 0.019 and 10 nM in assay buffer (50 mM Tris pH 7.4, 10 mM $MgCl_2$, 1 mM EDTA, with 150 mM NaCl or 150 mM KCl as needed) containing 0.1% protease-free BSA in a final volume of 250 µL per tube. Reactions were incubated at room temperature for 1 hr.

Non-specific binding was determined using reactions containing 10 mM theophylline. Reactions were separated on a vacuum manifold using GF/C filters (pre-soaked in assay +0.5% PEI) to retain membranes and discard unbound ligand. After washing four times with cold assay buffer, bound radioactivity was quantified using a scintillation counter. For competition binding experiments, aliquots of membranes were incubated with 0.5 nM [$^3$H]-ZM241385, and varying concentrations of cold NECA, from 0.5 nM to 300 µM, were included in the binding reactions. All binding experiments were carried out as three independent experiments, each performed in duplicate. Data analysis and fitting was performed with GraphPad Prism (GraphPad Software Inc.).

## Reconstitution and GTPγS binding

9 µg of $G_s$ trimer was added to a tube and preincubated on ice for 15 min with 75 µL of 1650 µM SAPE (1-stearoyl-2-arachidonoyl-$sn$-glycero-3-phosphoethanolamine; Avanti Lipids), 980 µM porcine brain phosphatidylserine (Avanti Lipids), 180 µM cholesteryl hemisuccinate (Steraloids) in 20 mM Hepes pH 8.0, 100 mM NaCl, 0.4% deoxycholate, 0.04% sodium cholate. 35 pmol of $A_{2A}$ receptor and HMEN buffer (20 mM HEPES pH 8.0, 100 mM NaCl, 3 mM $MgCl_2$, 1 mM EDTA) was added to the tube for a total volume of 150 µL. The sample was incubated an additional 5 min and applied to an Ultrogel AcA34 column (Sigma-Aldrich) equilibrated in HMEN buffer. BSA was added to fractions at a concentration of 0.1 mg/mL and vesicles were flash-frozen for storage at −80°C. Receptor recovery was monitored by total ZM241385 binding and ~1 mL after void volume was used in GTPγS binding assays. Recovery of receptor in this pool was ~13%.

5 µL of vesicles was assayed in a total of 50 µL with 20 mM HEPES pH 8.0, 100 mM NaCl, 2 mM $MgCl_2$, 1 mM EDTA, 1 mM DTT, 0.1 mg/mL BSA, 100 nM GTPγS with $^{35}$S GTPγS as a tracer in either no ligand, 10 µM NECA agonist, or 100 nM ZM241385 antagonist. Samples were incubated at 30°C for 10 min and stopped with 50 µL of quench buffer (20 mM Tris pH 8, 100 mM NaCl, 10 mM $MgCl_2$, 0.1 mM DTT, 1 mM GTP, and 0.1% Lubrol) and incubated on ice for 10 min. Samples were filtered over BA85 nitrocellulose and filters counted on a liquid scintillation counter after washing 4x with 20 mM Tris pH 8.0, 100 mM NaCl, 10 mM $MgCl_2$.

## Acknowledgements

This project was supported by the National Science Foundation (Grant 1000136529 to LC), the American Heart Association (Grant 16PRE27200004 to LC), the HRH Prince Alwaleed Bin Talal Bin Abdulaziz Alsaud Institute of Computational Biomedicine at Weill Medical College of Cornell University (GK and MVL), the Lundbeck Foundation (Grant R37-A3457 to SGFR), the Danish Independent Research Council (Grant 0602-02407B to SGFR), the Welch Foundation (I-1770 to DMR), a Packard Foundation Fellowship (to DMR), and the National Institutes of Health (R01GM113050 to DMR, R01GM106239 to KHG, R01MH054137 to GK). Reagents for NMR spectroscopy were generously provided by a Research Award from Cambridge Isotope Laboratories, Inc. Computational resources are gratefully acknowledged: (i) an allocation at the National Energy Research Scientific Computing Center (NERSC, repository m1710, used for carrying out MD simulations) supported by the Office of Science of the U.S. Department of Energy under Contract No. DE-AC02-05CH11231; and (ii) the David A Cofrin Center for Biomedical Information in the HRH Prince Alwaleed Bin Talal Bin Abdulaziz Alsaud Institute for Computational Biomedicine at Weill Cornell Medical College.

## Additional information

### Funding

| Funder | Grant reference number | Author |
| --- | --- | --- |
| Welch Foundation | I-1770 | Daniel M Rosenbaum |
| National Institutes of Health | R01GM113050 | Daniel M Rosenbaum |
| National Institutes of Health | R01GM106239 | Kevin H Gardner |
| National Science Foundation | 1000136529 | Lindsay D Clark |
| American Heart Association | 16PRE27200004 | Lindsay D Clark |

The funders had no role in study design, data collection and interpretation, or the decision to submit the work for publication.

## Author contributions

Lindsay D Clark, Igor Dikiy, Conceptualization, Investigation, Methodology, Writing—original draft, Writing—review and editing; Karen Chapman, Karin EJ Rödström, James Aramini, Michael V LeVine, George Khelashvili, Investigation, Methodology; Søren GF Rasmussen, Supervision, Investigation, Methodology, Writing—review and editing; Kevin H Gardner, Daniel M Rosenbaum, Conceptualization, Supervision, Funding acquisition, Investigation, Visualization, Methodology, Writing—original draft, Project administration, Writing—review and editing

## Author ORCIDs

Lindsay D Clark (iD) https://orcid.org/0000-0001-6879-6883
Igor Dikiy (iD) http://orcid.org/0000-0002-8155-7788
George Khelashvili (iD) https://orcid.org/0000-0001-7235-8579
Kevin H Gardner (iD) http://orcid.org/0000-0002-8671-2556
Daniel M Rosenbaum (iD) http://orcid.org/0000-0002-7174-4993

## Decision letter and Author response

Decision letter https://doi.org/10.7554/eLife.28505.027
Author response https://doi.org/10.7554/eLife.28505.028

# Additional files

## Supplementary files

• Transparent reporting form
DOI: https://doi.org/10.7554/eLife.28505.018

## Major datasets

The following dataset was generated:

| Author(s) | Year | Dataset title | Dataset URL | Database, license, and accessibility information |
|---|---|---|---|---|
| Lindsay D Clark, Igor Dikiy, Karen Chapman, James Aramini, Michael V LeVine, George Khelashvili, Søren GF Rasmussen, Kevin H Gardner, Daniel M Rosenbaum | 2017 | NMR spectroscopy data, including wild type and mutant spectra | http://www.bmrb.wisc.edu/ftp/pub/bmrb/entry_directories/bmr27261/ | Publicly available at the Biological Magnetic Resonance Data Bank (accession no. 27261) |

The following previously published datasets were used:

| Author(s) | Year | Dataset title | Dataset URL | Database, license, and accessibility information |
|---|---|---|---|---|
| Lebon G, Warne T, Edwards PC, Bennett K, Langmead CJ, Leslie AGW, Tate CG | 2011 | Thermostabilised HUMAN A2a Receptor with NECA bound | http://www.rcsb.org/pdb/explore.do?structureId=2ydv | Publicly available at the RCSB Protein Data Bank (Accession no: 2YDV) |
| Sharff AJ, Quiocho FA | 1993 | REFINED 1.8 ANGSTROMS STRUCTURE REVEALS THE MECHANISM OF BINDING OF A CYCLIC SUGAR, BETA-CYCLODEXTRIN, TO THE MALTODEXTRIN BINDING PROTEIN | http://www.rcsb.org/pdb/explore.do?structureId=1DMB | Publicly available at the RCSB Protein Data Bank (Accession no: 1DMB) |

| Liu W, Chun E, Thompson AA, Chubukov P, Xu F, Katritch V, Han GW, Heitman LH, Ijzerman AP, Cherezov V, Stevens RC, GPCR Network | 2012 | Crystal structure of the chimeric protein of A2aAR-BRIL in complex with ZM241385 at 1.8A resolution | http://www.rcsb.org/pdb/explore/explore.do?structureId=4EIY | Publicly available at the RCSB Protein Data Bank (Accession no: 4EIY) |

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
