## [Decision Letter]

Thank you for submitting your article "Ligand modulation of sidechain dynamics in a wild-type human GPCR" for consideration by *eLife*. Your article has been reviewed by three peer reviewers, and the evaluation has been overseen by a Reviewing Editor and Richard Aldrich as the Senior Editor. The following individual involved in review of your submission has agreed to reveal his identity: Scott Prosser (Reviewer #1).

The reviewers have discussed the reviews with one another and the Reviewing Editor has drafted this decision to help you prepare a revised submission.

Summary:

All reviewers agree that your work represents a significant methodological advance in the study of GPCR dynamics by NMR. There are, however, concerns about the sensitivity of the measurements and whether you are looking at 100% occupied states. The reviewers are also not convinced that the results as presented really advance understanding of how side chain dynamics are controlled by different classes of ligands.

Essential revisions:

1) This work takes an important step forward in the demonstration that (semi-) quantitative information can be obtained, but it is limited by sensitivity. This is apparent in the values of eta, per se, where very large errors are obtained. Can the authors think about additional experiments that might place their results on a more rigorous footing? Perhaps an additional dynamics measurement with a different agonist that is not as strong as NECA. Then one might predict less increase in dynamics. If the authors obtained this, it would provide additional support for their hypothesis. Alternatively, it is likely that carrying out ^13^C R_2_ experiments using ^13^CHD_2_ methyl groups (the appropriate precursor can be purchased) would provide significantly more sensitivity and if similar trends are observed then again this would provide more confidence in the results.

2) Ligand-dependent chemical shift changes: There is concern about both the lack of shifts in the inverse agonist versus agonist-saturated samples, as well as the robustness of the shifts suggested in Figure 3, which could just as easily arise from linewidth differences. Inspection of the spectra and examining the plot of the shifts in the graph leads to the impression that there is little correspondence between the two where apparently some peaks are shifting >10-fold more than others. It was also surprising that changes were not observed given the large conformation change observed in A_2A_R between the crystal structure of the ZM241385-bound state and the NECA-bound state. For these reasons, it is suggested that you make use of ROSETTA or some chemical shift prediction program and comment on the expected shifts between active and inactive states at least for the half-dozen or so Ile's that are removed from the hydrophobic interior and are somewhat closer to the solvent interface. There are also a few well-resolved Ile resonances that one might expect to be able to predict or confirm from ROSETTA.

3) It is unclear how reliable the assignments of amino acid residues really are. In one instance, mutating Ile274 to Met resulted in the loss of intensity of 4 peaks. The authors assume that 2 of them, having been assigned to other Ile residues, clearly do not belong Ile274 and that 'careful inspection' led to the assignment of one peak. It is unclear how the authors have done this. More importantly, how can the authors be certain that the other mutations that led to peak assignments did not also lead to loss of spectra in the unresolved peaks a-o and that these actually represent the Ile residues and not the peaks chosen? Also, why has the density of the 2 peaks at ~0.82 ppm 1H and 15-16 ppm ^[33]^C been ignored?

4) In Figure 1, the authors demonstrate that the perdeuterated A_2A_R couples to G_s_. However, the control experiment with the normal protonated A_2A_R is not done, so it is not possible to know whether the levels observed for the perdeuterated sample are significantly worse or not. In addition, the authors do not state the amount of functional receptor (as defined by radioligand binding) in each of the NMR samples. This is an issue given the non-optimal conditions for purification (see below).

5) The authors apparently exchange theophylline for NECA in a buffer containing 250 mM NaCl and then purified further on a SEC column containing 150 mM NaCl. NaCl is an allosteric antagonist of A_2A_R (EC50 of 40-50 mM), so how sure are the authors that in the NMR experiments using NECA that they have 100% ligand occupancy?

6) There are questions about why the A_2A_R was purified in DDM alone. A_2A_R is most stable in the presence of CHS and that if this is not added to the purification in DDM, then the receptor in the absence of ligand rapidly becomes inactive (Weiss and Grisshammer (2002) Eur. J. Biochem. 269, 82-92). Did the authors test the stability of their preparations by FSEC or radioligand binding to ascertain that there was no loss of functionality during NMR data collection? Moreover, it appears that functional purification was not attempted to remove non-functional receptor. If the authors do not use functional purification strategies they should comment on functional purity not so much in vesicles but in DDM. It would also be helpful to comment on how the use of DDM might influence dynamics.

7) The Abstract is bereft of substantive comments regarding the system under study. The authors should comment both on solvent exposure and relative effects with ligand/G protein and on the triple quantum based order parameter analysis. Something like "…we observe a diversity of ligand-dependent changes in η, with some Ile residues becoming more rigid with agonist while most become more flexible…". Even a comment on the negative results belongs in the Abstract in my opinion (in particular, the lack of chemical shift change with ligand).

---

## [Author Response]

Summary:All reviewers agree that your work represents a significant methodological advance in the study of GPCR dynamics by NMR. There are, however, concerns about the sensitivity of the measurements and whether you are looking at 100% occupied states. The reviewers are also not convinced that the results as presented really advance understanding of how side chain dynamics are controlled by different classes of ligands.

We appreciate the generally positive comments of the reviewers, and we also understand their concerns, both from the standpoint of ligand occupancy and advancing our understanding of ligand regulation of A_2A_R. As detailed below, we have tried to address these concerns through a combination of new data and analysis (i.e. new Figure 3 and Figure 6—figure supplement 1) as well as a modified Discussion.

Essential revisions:1) This work takes an important step forward in the demonstration that (semi-) quantitative information can be obtained, but it is limited by sensitivity. This is apparent in the values of eta, per se, where very large errors are obtained. Can the authors think about additional experiments that might place their results on a more rigorous footing? Perhaps an additional dynamics measurement with a different agonist that is not as strong as NECA. Then one might predict less increase in dynamics. If the authors obtained this, it would provide additional support for their hypothesis. Alternatively, it is likely that carrying out ^13^C R_2_ experiments using ^13^CHD_2_ methyl groups (the appropriate precursor can be purchased) would provide significantly more sensitivity and if similar trends are observed then again this would provide more confidence in the results.

We agree with the referees that the information we can obtain from our HMQC and 3Q experiments is limited by sensitivity. However, there are fundamental barriers to improving our signal/noise. As with many biologically significant integral membrane proteins, wild-type A_2A_R in detergent micelles presents a challenging target for NMR studies, given its size, flexibility, and our inability to increase concentration beyond 100 μM without precipitation. To address these issues, we invested substantial effort in developing protocols to generate and purify highly deuterated samples, which were exchanged into deuterated detergent to improve the S/N and dispersion of our spectra as needed to collect relaxation data such as the 3Q information here. Compared to studies using ^13^C-methyl labeling of Met sidechains in native GPCRs (i.e. Nygaard et al., Cell 2013, 152: 532), our S/N and dispersion is quite good. While we agree with the merits of the reviewers’ proposal to use ^13^CHD_2_ labeling to enhance our sensitivity in relaxation NMR experiments (something that we may now pursue with the practical expertise we have developed for this manuscript), the expense and time required for repeating our studies with this route is beyond the scope of the present work.

Given the relatively large errors in eta that we obtain from the 3Q relaxation data, we have focused on several routes to address these as we have revised this manuscript. To start, we have revisited the determination of these errors, now incorporating a Monte Carlo approach described in the Materials and methods. We have also included new molecular dynamics simulation data on A_2A_R bound to NECA and ZM241385 in the presence of Na^+^, which is shown in Figure 6—figure supplement 1 and detailed in the Materials and methods. Notably, the trends in S^2^ order parameters predicted by these simulations are in general agreement with the trends in eta that we observe in our experimental data, which helps validate our approach. From an experimental vantage, although we have not been able to purify mg quantities of A_2A_R with lower-affinity agonists such as adenosine, we have included new data on the effect of Na^+^ versus K^+^ on the Ile probes in A_2A_R (see new Figure 3 and response to comment 5 below). These newly acquired data, combined with our 3Q experiments in NaCl, show that low [Na^+^] is required for NECA to promote structural changes in the receptor, and that ZM241385 exerts a long-range allosteric effect to suppress dynamics at the G protein binding site. These experimental observations on wild-type A_2A_R are the first NMR data for sidechain probes in this receptor, and complement the A_2A_R crystal structures that were obtained for stabilizing mutants or fusion proteins.

2) Ligand-dependent chemical shift changes: There is concern about both the lack of shifts in the inverse agonist versus agonist-saturated samples, as well as the robustness of the shifts suggested in Figure 3, which could just as easily arise from linewidth differences. Inspection of the spectra and examining the plot of the shifts in the graph leads to the impression that there is little correspondence between the two where apparently some peaks are shifting >10-fold more than others. It was also surprising that changes were not observed given the large conformation change observed in A_2A_R between the crystal structure of the ZM241385-bound state and the NECA-bound state. For these reasons, it is suggested that you make use of ROSETTA or some chemical shift prediction program and comment on the expected shifts between active and inactive states at least for the half-dozen or so Ile's that are removed from the hydrophobic interior and are somewhat closer to the solvent interface. There are also a few well-resolved Ile resonances that one might expect to be able to predict or confirm from ROSETTA.

The chemical shift changes between the NECA- and ZM241385-bound A_2A_R are indeed small, and we acknowledge the valid criticism that these shifts could be due to linewidth differences. Therefore we have removed previous Figure 3, which contained a plot of the changes. On the other hand, new NMR data for the receptor in the absence of Na^+^ indicate larger ligand-dependent changes in chemical shift and broadening for the NECA-bound receptor (new Figure 3 and response to comment 5). Comparison of crystal structures of engineered A_2A_R constructs bound to ZM241385 (i.e. Liu et al., Science 2012, 337: 232) and NECA (Lebon et al., Nature 2011, 474: 521) do show significant changes, with the agonist-bound structure representing an “active-like intermediate” conformation. These changes lead to disruption and collapse of the Na^+^ pocket in A_2A_R, and occur despite the presence of Na^+^ above its EC_50_ in the crystallization experiments. Our data suggest that Na^+^ must be at a low concentration for NECA to stabilize an altered conformation of the receptor, and the crystal structures may have captured such an altered conformation due to the use of thermostabilizing mutations or fusion proteins. Despite the very small chemical shift changes between NECA- and ZM241385-bound receptors seen in our spectra with Na^+^ present, we were still able to differentiate these samples by differences in their 3Q relaxation profiles, which we believe relate to the mechanism of ZM241385’s inverse agonism, as described in the Discussion.

In terms of chemical shift prediction, we have used the CH3SHIFT program from Michele Vendruscolo’s group (Sahakyan et al., J. Biomol. NMR2011, 50: 331), taking advantage of its strengths in sidechain chemical shift calculations (as opposed to most other programs, which focus solely on backbone shifts). While we have been able to use this program to predict the ^1^H shifts of Ile δ1 methyl groups in the active and inactive conformations of the MD simulations, the ligand-induced changes observed here are modest in magnitude and below the ~0.2 ppm stated accuracy of the program. Combined with the inability to generate ^13^C predictions, we have unfortunately found that this approach is not robust enough to give useful insights at this time.

3) It is unclear how reliable the assignments of amino acid residues really are. In one instance, mutating Ile274 to Met resulted in the loss of intensity of 4 peaks. The authors assume that 2 of them, having been assigned to other Ile residues, clearly do not belong Ile274 and that 'careful inspection' led to the assignment of one peak. It is unclear how the authors have done this. More importantly, how can the authors be certain that the other mutations that led to peak assignments did not also lead to loss of spectra in the unresolved peaks a-o and that these actually represent the Ile residues and not the peaks chosen? Also, why has the density of the 2 peaks at ~0.82 ppm 1H and 15-16 ppm ^13^C been ignored?

Assignment of peaks for these samples was extremely difficult due to the fundamental limitation of the “assignment by mutagenesis” approach: the potential for mutations to introduce wide-ranging changes in the structure and dynamics of different regions of the receptor, causing broadening or doubling of multiple additional peaks. As such, we had to use multiple sources of data to make reasonable assignments. In the revised manuscript, we have laid out the logic and details of these assignments more thoroughly. The I274M mutant is an example of a mutant that was difficult to use alone for assignment, due to such bystander effects. As described, we used a combination of the I274V and I274M spectra, in addition to solvent PRE data, to assign the peak at 0.55 ppm/14.1 ppm (^1^H/^13^C) to this residue. We have also added an additional panel in Figure 2—figure supplement 2 that identifies the peaks at 0.82ppm/15-16ppm (^1^H/^13^C) as buffer components.

4) In Figure 1, the authors demonstrate that the perdeuterated A_2A_R couples to Gs. However, the control experiment with the normal protonated A_2A_R is not done, so it is not possible to know whether the levels observed for the perdeuterated sample are significantly worse or not. In addition, the authors do not state the amount of functional receptor (as defined by radioligand binding) in each of the NMR samples. This is an issue given the non-optimal conditions for purification (see below).

We have now included the control experiment with reconstituted protonated A_2A_R in Figure 1, which shows a similar level of stimulation by NECA and a similar level of inverse agonism by ZM241385 compared to the perdeuterated receptor. We have not carried out radioligand binding on the purified detergent-solubilized receptor in our NMR samples, since they are exchanged into buffer with a saturating amount of high-affinity ligand during IMAC and SEC purification steps. However we believe that we have a high percent of functional ligand-bound receptor, as described below in response to comment 5.

5) The authors apparently exchange theophylline for NECA in a buffer containing 250 mM NaCl and then purified further on a SEC column containing 150 mM NaCl. NaCl is an allosteric antagonist of A_2A_R (EC50 of 40-50 mM), so how sure are the authors that in the NMR experiments using NECA that they have 100% ligand occupancy?

We are grateful to the reviewers for pointing out that our previous data was all collected in 150 mM NaCl, and that Na^+^ acts as a negative allosteric modulator of A_2A_R. As a result of this comment, we carried out additional experiments on NECA- and ZM241385-bound A_2A_R in the presence of 150 mM KCl instead of NaCl. This data is shown in the new Figure 3, and described in the Results and Discussion section. Our overall conclusion is that low [Na^+^] is required for NECA binding to stabilize significant structural changes in A_2A_R that can be measured with our Ile probes distributed throughout the receptor. This dependency is different than what is implied by engineered A_2A_R crystal structures, in which NECA binding is sufficient to overcome the presence of high [Na^+^] (used at concentrations above its EC_50_) and drive structural changes that alter the core near I92^3.40^ and I238^6.40^.

In terms of NECA occupancy, we have several lines of evidence that this value is high. First, our measured K_i_ of 4 μM NECA (in competition with ^3^H-ZM241385) in the presence of 150 mM NaCl is still well below the concentration of 20 μM we used in our NMR experiments. Second, the spectrum of NECA in Na^+^ is clearly differentiated from the apo spectrum (Figure 2). Given that theophylline has a low affinity (K_i_ ~ 3 mM) and a corresponding fast off-rate, our extensive washing with saturating NECA during IMAC and SEC should be sufficient to fully exchange the ligand. Finally, we showed that our purified DDM-solubilized A_2A_R exchanged into NECA (in NaCl buffer) binds to G_s_ heterotrimer with extremely high efficiency, leaving no free receptor (Figure 2—figure supplement 1). Together these data give us confidence that the spectrum in Figure 2 represents A_2A_R with near-100% NECA occupancy.

6) There are questions about why the A_2A_R was purified in DDM alone. A_2A_R is most stable in the presence of CHS and that if this is not added to the purification in DDM, then the receptor in the absence of ligand rapidly becomes inactive (Weiss and Grisshammer (2002) Eur. J. Biochem. 269, 82-92). Did the authors test the stability of their preparations by FSEC or radioligand binding to ascertain that there was no loss of functionality during NMR data collection? Moreover, it appears that functional purification was not attempted to remove non-functional receptor. If the authors do not use functional purification strategies they should comment on functional purity not so much in vesicles but in DDM. It would also be helpful to comment on how the use of DDM might influence dynamics.

Our initial purification strategy for A_2A_R followed previously published protocols and included CHS as part of the micelle composition, however we encountered significant detergent artifact peaks in the Ileδ1 methyl region of our 2D HMQC spectra that drowned out the signals arising from the methyl peaks of our labeled receptor. These artifacts effectively prevented us from carrying out relaxation experiments. Our system also required thorough exchange from protonated DDM into deuterated DDM to reduce spectral artifacts. While we appreciate that A_2A_R may be less stable without CHS in the absence of ligands, as previously reported, most of our studies were carried out in the presence of saturating concentrations of NECA or ZM241385. Indeed the stability of the apo receptor was the poorest of all of our samples (in NaCl buffer), judging by the loss of material upon concentration and precipitation during data collection at 30 °C.

Functional purification by ligand-affinity chromatography (XAC column) was used previously in a pioneering study on A_2A_R dynamics by ^19F^ NMR (Ye et al., Nature 2016, 533: 365). On the other hand, multiple crystal structures of A_2A_R (including wild-type receptor-fusion protein chimeras) bound to different ligands have used purification protocols similar to ours, with only IMAC and SEC steps. In the present work, we were not able to apply a ligand-affinity chromatography step to our samples. Despite not using CHS or ligand-affinity chromatography, we still observed highly efficient complex formation between our NECA-bound receptor and G_s_ heterotrimer, both purified in DDM (Figure 2—figure supplement 1). We did not carry out solution radioligand binding before and after NMR data acquisition, due to the presence of saturating concentrations of high-affinity ligands in our samples. Some sample precipitation was typically observed during extended incubation of A_2A_R at 30 °C. We did assess sample stability by collecting short HMQC spectra initially and comparing subsequent spectra over the period of data acquisition. A side-by-side example of NECA-bound A_2A_R before and after 7.5 hr at 30 °C is shown in Figure 2—figure supplement 3. Despite some loss of signal (presumably due to protein precipitation), these spectra still show the same peaks and chemical shifts.

In the present study, we have shown that it is possible to collect methyl TROSY spectra and carry out dynamics experiments on a wild-type GPCR in a DDM micelle with an aggregate molecular weight of ~100-150kDa. We acknowledge that even though we have demonstrated receptor functionality in DDM micelles (i.e. ability to efficiently bind G protein), this is not an ideal membrane mimetic. Our experimental approach will enable future studies of sidechain dynamics in wild-type GPCRs reconstituted into lipid nanodiscs or SMALPs, where receptor perdeuteration should facilitate data acquisition despite the large size of these particles.

7) The Abstract is bereft of substantive comments regarding the system under study. The authors should comment both on solvent exposure and relative effects with ligand/G protein and on the triple quantum based order parameter analysis. Something like "…we observe a diversity of ligand-dependent changes in η, with some Ile residues becoming more rigid with agonist while most become more flexible…". Even a comment on the negative results belongs in the Abstract in my opinion (in particular, the lack of chemical shift change with ligand).

We have now re-written the Abstract to focus on how this study contributes to our understanding of the biophysical properties of A_2A_R, rather than the methodological advances.